# MGDA Converges under Generalized Smoothness, Provably

**Qi Zhang**[*1]**, Peiyao Xiao**[*2]**, Shaofeng Zou**[1] **& Kaiyi Ji**[2]

School of Electrical, Computer and Energy Engineering, Arizona State University[1]
Department of Computer Science and Engineering, University at Buffalo[2]
{qzhan261,zou}@asu.edu, {peiyaoxi,kaiyiji}@buffalo.edu

## Abstract

Multi-objective optimization (MOO) is receiving more attention in various fields such as multi-task learning. Recent works provide some effective algorithms with theoretical analysis but they are limited by the standard $L$-smooth or bounded-gradient assumptions, which typically do not hold for neural networks, such as Long short-term memory (LSTM) models and Transformers. In this paper, we study a more general and realistic class of generalized $\ell$-smooth loss functions, where $\ell$ is a general non-decreasing function of gradient norm. We revisit and analyze the fundamental multiple gradient descent algorithm (MGDA) and its stochastic version with double sampling for solving the generalized $\ell$-smooth MOO problems, which approximate the conflict-avoidant (CA) direction that maximizes the minimum improvement among objectives. We provide a comprehensive convergence analysis of these algorithms and show that they converge to an $\epsilon$-accurate Pareto stationary point with a guaranteed $\epsilon$-level average CA distance (i.e., the gap between the updating direction and the CA direction) over all iterations, where totally $\mathcal{O}(\epsilon^{-2})$ and $\mathcal{O}(\epsilon^{-4})$ samples are needed for deterministic and stochastic settings, respectively. We prove that they can also guarantee a tighter $\epsilon$-level CA distance in each iteration using more samples. Moreover, we analyze an efficient variant of MGDA named MGDA-FA using only $\mathcal{O}(1)$ time and space, while achieving the same performance guarantee as MGDA.

## 1 Introduction

There have been a variety of emerging applications of multi-objective optimization (MOO), such as online advertising (Ma et al., 2018), autonomous driving (Huang et al., 2019), and reinforcement learning (Thomas et al., 2021). Mathematically, the MOO problem takes the following formulation.

$$F^* = \min_{x \in \mathbb{R}^m} F(x) := (f_1(x), f_2(x), ..., f_K(x)), \tag{1}$$

where $K$ is the total number of objectives and $f_k(x)$ is the $k$-objective function given model parameters $x$. Under the stochastic setting, $f_k(x) = \mathbb{E}_s[f_k(x; s)]$, where $s$ denotes data sample. In MOO setting, we are interested in optimizing all the objectives simultaneously. However, this problem is challenging due to the gradient conflict that some objectives with larger gradients dominate the update direction at the sacrifice of significant performance degeneration on the less-fortune objectives with smaller gradients. Thus, a widely-adopted target is to find the Pareto stationary point $x$ that the performance of all objectives cannot be further improved without compromising some objectives. A variety of MOO-based methods have been proposed to mitigate this conflict and find a more balanced solution among all objectives. In particular, the multiple gradient descent algorithm (MGDA) (Désidéri, 2012) aims to find a conflict-avoidant (CA) update direction that maximizes the minimal improvement among all objectives and converges to a Pareto stationary point at which there is no common descent direction for all objective functions. This idea then inspired numerous follow-up methods including but not limited to CAGrad (Liu et al., 2021), PCGrad (Yu et al., 2020), GradDrop (Chen et al., 2020), FAMO (Liu et al., 2024) and FairGrad (Ban & Ji, 2024) with a

---

*Equal contribution.

convergence guarantee in the deterministic setting with full-gradient computations. The theoretical understanding of the convergence and complexity of stochastic MOO is not well-developed until very recently. Liu & Vicente (2021) proposed stochastic multi-gradient (SMG) as a stochastic version of MGDA, and established its convergence guarantee. Zhou et al. (2022) analyzed the non-convergence issues of MGDA, CAGrad and PCGrad in the stochastic setting, and further proposed a convergent approach named CR-MOGM. More recently, Fernando et al. (2022) and Chen et al. (2024) proposed single-loop stochastic MOO methods named MoCo and MoDo, and proved their convergence to an $\epsilon$-accurate Pareto stationary point while guaranteeing an $\epsilon$-level **average CA distance**[1] over all iterations. Xiao et al. (2024) proposed a double-loop algorithm named SDMGrad that enables to obtain an unbiased stochastic multi-gradient via a double-sampling strategy. They established the convergence of SDMGrad with a guaranteed $\epsilon$-level CA distance in every iteration, which we call as **iteration-wise CA distance**.

However, all existing works are limited by the standard $L$-smooth and bounded-gradient assumptions. Nevertheless, a recent study (Zhang et al., 2019) indicates that such assumptions may not necessarily be true for the training of neural networks and an alternative $(L_0, L_1)$-smoothness condition was observed and studied, which assumes the Lipschitz constant to be linear in the gradient norm and the gradient norm to be potentially infinite. Furthermore, this phenomenon has been consistently observed in our experiments (e.g., see Figure 1). Interestingly, it has been widely observed that MGDA algorithms always converge even under such generalized smoothness conditions (Sener & Koltun, 2018; Liu et al., 2021; Xiao et al., 2024). This naturally raises a thought-provoking question:

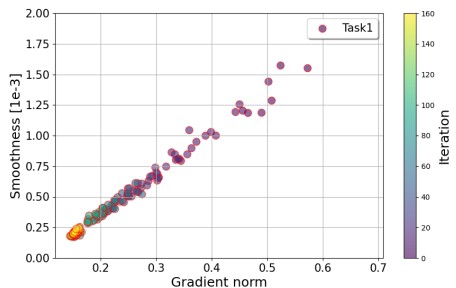

Figure 1: Local smoothness constant vs. gradient norm for training SegNet on CityScapes dataset of Task 1.

Q. *Can the fundamental MGDA algorithm provably converge under the generalized smoothness condition, while achieving a sufficiently small CA distance?*

This question remains open due to the **following challenges**. The analysis of existing MOO methods cannot be generalized to this $(L_0, L_1)$-smoothness directly due to the possible unbounded smoothness or gradient norm. In addition, all existing works (Reisizadeh et al., 2023; Zhang et al., 2019; Li et al., 2024b;a; Jin et al., 2021; Crawshaw et al., 2022; Chen et al., 2023; Zhang et al., 2024) in generalized smoothness are limited to the single task problems, which are fundamentally different from the MOO problems since even though each single task is generalized smooth, the linear combination of these tasks is not necessarily generalized smooth. In this paper, we provide an affirmative answer to this question. Our main contributions are summarized below.

## 1.1 OUR CONTRIBUTIONS

We establish a comprehensive convergence analysis for MGDA under the generalized smoothness condition, in both the deterministic and stochastic settings. Moreover, the analysis covers both the average CA distance and the iteration-wise CA distance.

**Weakest assumptions in MOO.** In this paper, we investigate the generalized $\ell$-smooth assumption, where $\ell$ is a general non-decreasing function of gradient norm, and includes both the standard $L$-smooth and $(L_0, L_1)$-smooth assumptions as special cases. This assumption finds many applications, such as LSTM models (Zhang et al., 2019), transformers (Crawshaw et al., 2022), distributionally robust optimization (Jin et al., 2021) and higher-order polynomial functions (Chen et al., 2023). In addition, we **do not make any bounded-gradient assumption**, which is required in previous analysis to ensure the bounded multi-gradient approximation. To the best of our knowledge, this is the first work to investigate generalized smoothness in MOO problems. Table 1 presents a detailed comparison of assumptions with existing analyses.

**Convergence analysis.** We first show that the vanilla MGDA method can converge to an $\epsilon$-accurate Pareto stationary point, while guaranteeing a small $\epsilon$-level average CA distance. A warm start process

---

[1]CA distance means the distance between the updating direction and the CA direction. Its formal definition can be found in Section 2.4

is introduced before the main loop of MGDA to achieve a more aggressive $\epsilon$-level iteration-wise CA distance. In the stochastic setting, we provide the convergence guarantee for MGDA with a double sampling, which was introduced by Xiao et al. (2024); Chen et al. (2024) to obtain unbiased multi-gradient approximations. Furthermore, we analyze a computation- and memory-efficient variant of MGDA, named MGDA with fast approximation (MGDA-FA), which updates the objective weights $w$ using only forward passes of $F(\cdot)$ rather than gradient $\nabla F$, effectively reducing $O(K)$ time and space to $O(1)$ without hurting the performance guarantee.

**Sample complexity comparison.** To achieve an $\epsilon$-accurate Pareto stationary point and an $\epsilon$-level average CA distance, we show that MGDA require $\mathcal{O}(\epsilon^{-2})$ and $\mathcal{O}(\epsilon^{-4})$ samples in the deterministic and stochastic settings, respectively. **Both of these complexities match with the existing best results.** Furthermore, to achieve an $\epsilon$-level iteration-wise CA distance, MGDA with warm start requires an increased number of samples, at the order of $\mathcal{O}(\epsilon^{-11})$ and $\mathcal{O}(\epsilon^{-17})$ respectively, in both deterministic and stochastic scenarios, due to smaller step sizes and mini-batch data sampling, shown in Table 1. Typically, achieving an $\epsilon$-**level iteration-wise CA** distance results in much higher sample complexity, such as $\mathcal{O}(\epsilon^{-24})$ in Fernando et al. (2022), $\mathcal{O}(\epsilon^{-12})^2$ in Chen et al. (2024) and $\mathcal{O}(\epsilon^{-12})$ in Xiao et al. (2024) for non-convex stochastic setting. Moreover, we show that MGDA-FA achieves the same performance guarantee as vanilla MGDA.

## 1.2 RELATED WORKS

**Gradient-based multi-objective optimization.** A variety of gradient manipulation techniques have emerged for simultaneous learning of multiple tasks. One prevalent category of methods adjusts the weights of various objectives according to factors such as uncertainty (Kendall et al., 2018), gradient norm (Chen et al., 2018), and training complexity (Guo et al., 2018). Methods based on MOO have garnered increased attention due to their systematic designs, enhanced training stability, and model-agnostic nature. For instance, Sener & Koltun (2018) framed Multi-Task Learning (MTL) as a MOO problem and introduced an optimization method akin to MGDA (Désidéri, 2012). Afterward, many MGDA-based methods have been proposed to mitigate gradient conflict with promising empirical performance. Among them, PCGrad (Yu et al., 2020) avoids conflict by projecting the gradient of each task on the norm plane of other tasks. GradDrop (Chen et al., 2020) randomly drops out conflicted gradients. CAGrad (Liu et al., 2021) adds a constraint on the update direction to be close to the average gradient. NashMTL (Navon et al., 2022) and FairGrad (Ban & Ji, 2024) formulated MTL as a bargaining game and a resource allocation problem, respectively. Theoretically, Fernando et al. (2022) proposed a provably convergent stochastic MOO method named MoCo based on an auxiliary tracking variable for gradient approximation. Chen et al. (2024) characterized the trade-off among optimization, generalization, and conflict avoidance in MOO. Xiao et al. (2024) proposed and analyzed a stochastic MOO method named SDMGrad with a preference-oriented regularizer. However, all these works rely on the $L$-smoothness and bounded-gradient assumptions. In contrast, this paper focuses on the MOO problems with generalized $\ell$-smooth objectives.

**Generalized smoothness.** The generalized $(L_0, L_1)$-smoothness was firstly proposed by Zhang et al. (2019), which was observed from extensive empirical experiments in training neural networks. A clipping algorithm was developed by Zhang et al. (2019) and the convergence rate was provided. Later, Jin et al. (2021) analyzed the convergence of a normalized momentum method. The SPIDER algorithm was also applied to solve generalized smooth problems in Reisizadeh et al. (2023); Chen et al. (2023), where Chen et al. (2023) studied a new notion of $\alpha$-symmetric generalized smoothness, which includes $(L_0, L_1)$-smoothness as a special case. Very recently, a new generalized $\ell$-smoothness condition was studied in Li et al. (2024a;b), which is the weakest smoothness condition and includes all the smoothness conditions discussed above. However, all the existing works on generalized smoothness are limited to single-task optimizations and the understanding of MOO is insufficient. This paper provides the first study of MOO under the generalized $\ell$-smoothness condition.

## 2 PRELIMINARIES

### 2.1 GENERALIZED SMOOTHNESS

The standard $L$-smoothness condition is widely investigated in existing optimization studies (Ghadimi & Lan, 2013; Ghadimi et al., 2016), which assumes a function $f : \mathcal{X} \to \mathbb{R}$ to be $L$-smooth if

---

[2]The order differs from (Chen et al., 2024) due to different definitions of the $\epsilon$-accurate Pareto stationarity and is taken when both $\epsilon$-accurate Pareto stationarity and $\epsilon$-level iteration-wise CA distance can be achieved.

| Method | Smoothness [1] | Assumption [2] | Setting | Complexity [3] |
|---|---|---|---|---|
| CAGrad (Liu et al., 2021) | (LS) | (OD) | Deterministic | N/A |
| PCGrad (Yu et al., 2020) | (LS) | (C) or (BC) | Deterministic | N/A |
| SMG (Liu & Vicente, 2021) | (LS) | (C) or (BC) | Stochastic | N/A |
| CR-MOGM (Zhou et al., 2022) | (LS) | (BF) and (BG) | Stochastic | $\mathcal{O}(\epsilon^{-4})$ |
| MoCo (Fernando et al., 2022) | (LS) | (BF) and (BG) | Stochastic | $\mathcal{O}(\epsilon^{-4})$ |
| MoDo (Chen et al., 2024) | (LS) | (BG) | Stochastic | $\mathcal{O}(\epsilon^{-4})$ |
| SDMGrad (Xiao et al., 2024) | (LS) | (BG) | Stochastic | $\mathcal{O}(\epsilon^{-4})$ |
| **MGDA (Them. 1 and 3 )** | **(GS)** | **N/A** | **Deterministic** | $\mathcal{O}(\epsilon^{-2})$ |
| **Stochastic MGDA (Them. 2 and 6)** | **(GS)** | **N/A** | **Stochastic** | $\mathcal{O}(\epsilon^{-4})$ |

Table 1: Comparison to assumptions in existing analyses. MGDA (Désidéri, 2012) assumes the access of optimal update direction and step size for each iteration and gets an asymptotic result thus is omitted in the table. Explanation on the upper footmarks: 1 : (LS) indicates that the objectives are standard $L$-smooth while (GS) the objectives are generalized $\ell$-smooth as defined in Definition 1; 2 : (OD) denotes the assumption of optimal direction in each iteration, (C) denotes the convex loss function assumption, (BC) denotes a lower bound on multi-task curvature, which is defined as $H(f, x, x') = \int_0^1 \nabla f(x)^\top \nabla^2 f(x + a(x - x')) \nabla f(x) da$ for a function $f$, (BF) denotes the bounded function value assumption and (BG) denotes the bounded gradient assumption; 3 : Sample complexity achieving $\epsilon$-accurate Pareto stationary point. N/A denotes no exploration on non-convex settings.

there exists a bounded constant $L$ such that for any $x, y \in \mathcal{X}$, $\|\nabla f(x) - \nabla f(y)\| \le L\|x - y\|$. Nevertheless, recent studies show that in the training of neural networks such as LSTM models (Zhang et al., 2019), transformers (Crawshaw et al., 2022), distributionally robust optimization (Jin et al., 2021) and high-order polynomials functions (Chen et al., 2023), the standard $L$-smoothness assumption does not hold. Instead, a generalized $(L_0, L_1)$-smoothness assumption was observed and studied in the training of LSTM models in Zhang et al. (2019), which assumes that for any $x \in \mathcal{X}$, $\|\nabla^2 f(x)\| \le L_0 + L_1\|\nabla f(x)\|$. This assumption implies the Lipschitz constant is potentially unbounded and reduces to the $L$-smoothness if $L_1 = 0$. Later, a more generalized assumption was proposed and studied in Li et al. (2024a):

**Definition 1.** *(Generalized $\ell$-smoothness, Definition 1 in Li et al. (2024a)). A real-valued differentiable function $f : \mathcal{X} \to \mathbb{R}$ is generalized $\ell$-smooth if $\|\nabla^2 f(x)\| \le \ell(\|\nabla f(x)\|)$ almost everywhere in $\mathcal{X}$,* **where $\ell : [0, +\infty) \to (0, +\infty)$ is a continuous non-decreasing function.**

The $(L_0, L_1)$-smoothness is a special case of generalized $\ell$-smoothness, where $\ell(a) = L_0 + L_1 a$. Another definition of generalized smooth is widely used and equivalent to the $\ell$-smoothness:

**Definition 2.** *($(r, \ell)$-smoothness, Definition 2 in Li et al. (2024a)). A real-valued differentiable function $f : \mathcal{X} \to \mathbb{R}$ is $(r, \ell)$-smooth if 1) for any $x \in \mathcal{X}, B(x, r(\|\nabla f(x)\|)) \in \mathcal{X}$, and 2) for any $x_1, x_2 \in B(x, r(\|\nabla f(x)\|))$, $\|\nabla f(x_1) - \nabla f(x_2)\| \le \ell(\|\nabla f(x)\|)\|x_1 - x_2\|$, where for continuous functions $r, \ell : [0, +\infty) \to (0, +\infty)$, $r$ is non-increasing, $\ell$ is non-decreasing and $B(x, R)$ is the Euclidean ball centered at $x$ with radius $R$.*

In $B(x, r(\|\nabla f(x)\|))$, $f$ is also $L$-smooth where $L = \ell(\|\nabla f(x)\|)$. Definitions 1 and 2 are equivalent (Li et al., 2024a): An $(r, \ell)$-smooth function is $\ell$-smooth; and a $\ell$-smooth function satisfying Assumption 1 is $(r, m)$-smooth with $m(u) := \ell(u + a)$ and $r(u) := a/m(u)$ for any $a > 0$.

## 2.2 PARETO CONCEPTS IN MULTI-OBJECTIVE OPTIMIZATION (MOO)

MOO aims to find points at which there is no common descent direction for all objectives. Considering points $x_1, x_2 \in \mathbb{R}^m$, we claim that $x_1$ dominates $x_2$ if $f_i(x_1) \ge f_i(x_2)$ for all $i \in [K]$ and $F(x_1) \ne F(x_2)$. We say a point is *Pareto optimal* if it is not dominated by any other point. In other words, we cannot improve one objective without compromising another when we reach a *Pareto optimal point*. In non-convex settings, MOO aims to find a *Pareto stationary point* defined as follows.

**Definition 3.** $x \in \mathbb{R}^m$ *is a Pareto stationary point if $\min_{w \in \mathcal{W}} \|\nabla F(x)w\|^2 = 0$, where $\mathcal{W}$ is the probability simplex over $[K]$. $x$ an $\epsilon$-accurate Pareto stationary point if $\min_{w \in \mathcal{W}} \|\nabla F(x)w\|^2 \le \epsilon^2$.*

## 2.3 Existing MOO Algorithms

**Deterministic MOO.** One of the big challenges of MOO is the gradient conflict, i.e., the gradients of different objectives may vary heavily in scale such that the largest gradient dominates the update direction. As a result, the performance of those objectives with smaller gradients (Yu et al., 2020) may be significantly compromised. As the most fundamental MOO algorithm, MGDA tends to find a balanced update direction for all objectives by considering the minimum improvement across all objectives and maximizes it by solving the following problem

$$\max_{d \in \mathbb{R}^m} \min_{i \in [K]} \left\{ \frac{1}{\alpha} (f_i(x) - f_i(x - \alpha d)) \right\} \approx \max_{d \in \mathbb{R}^m} \min_{i \in [K]} \langle \nabla f_i(x), d \rangle, \tag{2}$$

where $\alpha$ is the step size, $d$ is the update direction, and the first-order Taylor approximation is applied at $x$. To efficiently solve the above problem in eq. (2), we substitute the following relation

$$\max_{d \in \mathbb{R}^m} \min_{i \in [K]} \langle \nabla f_i(x), d \rangle - \frac{1}{2} \|d\|^2 = \max_{d \in \mathbb{R}^m} \min_{w \in \mathcal{W}} \left\langle \sum_{k=1}^{K} \nabla f_i(x) w_i, d \right\rangle - \frac{1}{2} \|d\|^2, \tag{3}$$

where the regularization term $-\frac{1}{2}\|d\|^2$ is to regulate the magnitude of our update direction. The solution to the problem in eq. (3) can be obtained by solving the following problem (Désidéri, 2012):

$$d^* = \nabla F(x) w^*; \quad s.t. \ w^* \in \arg\min_{w \in \mathcal{W}} \frac{1}{2} \|\nabla F(x) w\|^2. \tag{4}$$

The above approach has been widely used in various variants of MGDA such as SDMGrad, CAGrad, and PCGrad (Xiao et al., 2024; Yu et al., 2020; Liu et al., 2021).

**Stochastic MOO.** SMG (Liu & Vicente, 2021) is the first stochastic MGDA. It directly replaces the gradients with stochastic gradients and the update rule becomes

$$d_s^* = \nabla F(x; s) w_s^*; \quad s.t. \ w_s^* \in \arg\min_{w \in \mathcal{W}} \frac{1}{2} \|\nabla F(x; s) w\|^2,$$

where $\nabla F(x; s)$ is the estimate of $\nabla F(x)$ based on the sample $s$. However, this leads to a **biased** gradient estimation of the update direction $d_s^*$, and thus it requires an increasing batch size. To solve this issue, another work MoCo (Fernando et al., 2022) introduces a tracking variable $Y$ as a stochastic estimation of the true gradient. Afterward, a double-sampling strategy is proposed by Chen et al. (2024); Xiao et al. (2024) to generate a near-unbiased update direction.

**Strong assumptions in analysis.** All the works mentioned above require bounded gradients such as Chen et al. (2024); Fernando et al. (2022); Xiao et al. (2024) or $L$-smoothness such as Ban & Ji (2024); Liu et al. (2021); Navon et al. (2022); Yang et al. (2024). Their analyses do not apply to generalized $\ell$-smoothness objectives, since the Lipschitz constant is potentially infinity.

## 2.4 Conflict-avoidant (CA) direction and CA distance

We call the update direction $d^*$ in eq. (4) the *conflict-avoidant* (CA) direction since it mitigates gradient conflict. Though it may not be feasible to calculate the exact CA direction, we aim to find an update direction to be close to the CA direction. Therefore, measuring the gap between the CA direction and the estimated update direction is important, which we define as the CA distance.

**Definition 4.** $\|d - d^*\|$ *is the CA distance between estimated update direction $d$ and CA direction $d^*$.*

The larger the CA distance is, the further the estimated update direction will be away from the CA direction, and the more conflict there will be. In single-loop algorithms, MoCo (Fernando et al., 2022) ensures the average CA distance over iteration is of the order of $\epsilon$, while MoDo (Chen et al., 2024) guarantees the $\epsilon$-order iteration-wise CA weight distance (as stated in their Theorem 3.4). Meanwhile, the double-loop algorithm SDMGrad (Xiao et al., 2024) guarantees an $\epsilon$-order CA distance in every iteration. In this work, we analyze the CA distance in both cases and provide convergence results.

## 3 MGDA Algorithms under Generalized Smoothness

### 3.1 MGDA with and without Warm Start

It has been shown in eq. (4) that MGDA needs to approximate the optimal weight $w^*$ and the optimal updating direction $d^*$. However, since the optimal weight $w^*$ of the convex function is not unique,

we deal with this issue by adding an $\ell_2$ regularization term and the problem becomes

$$w_\rho^* = \arg \min_{w \in \mathcal{W}} \frac{1}{2}\|\nabla F(x)w\|^2 + \frac{\rho}{2}\|w\|^2. \tag{5}$$

Besides the benefit of a unique solution, adding an $\ell_2$ regularization term also makes $w_\rho^*(x)$ Lipschitz continuous (Fernando et al., 2022). Note that $w^*(x)$ may not be Lipschitz continuous because $\nabla F(x)^\top \nabla F(x)$ may not be positive definite. Nevertheless, the analysis of CA distance is difficult because $w^*$ may not be Lipschitz continuous. Thus, we will characterize the gap between $w^*$ and $w_\rho^*$ plus the change of $w_\rho^*$ after adding this $\ell_2$ regularization term. As a result, the update rules become Lines 5-6 in Algorithm 1. We first update $w_t$ by a projected gradient descent process and compute the update direction $d_t = \nabla F(x_t)w_t$ to update model parameters.

For our single-loop algorithm, CA distance is proportional to the term $\|w_t - w_{t,\rho}^*\|$, which decreases as the algorithm iterates with some error terms controlled by appropriately chosen small step sizes. If we initialize $w_0$ randomly, $\|w_0 - w_{0,\rho}^*\|$ will be a constant order, and so will the first CA distance. Meanwhile, we can only get an $\epsilon$-order CA distance after a certain iteration number $t' > 1$ when $\|w_{t'} - w_{t',\rho}^*\|$ takes an $\epsilon$ order. Thus, we introduce an extra warm start process using Algorithm 2 to guarantee the new $w_0$ is close enough to $w_{0,\rho}^*$ and a small level CA distance in every iteration. **However, this warm start process is not needed if we only require a small averaged CA distance.**

---

**Algorithm 1** Single loop MGDA with and without warm start

---
1: **Initialize:** model parameters $x_0$, weights $w_0$ and a constant $\rho$
2: *Option I:* $w_0$=**warm-start**$(w_0, x_0, \rho)$    # for analyzing iteration-wise CA distance
3: *Option II:* $w_0 \leftarrow w_0$                # for analyzing averaged CA distance
4: **for** $t = 0, 1, ..., T-1$ **do**
5:      $w_{t+1} = \Pi_{\mathcal{W}}\left(w_t - \beta[\nabla F(x_t)^\top \nabla F(x_t)w_t + \rho w_t]\right)$
6:      $x_{t+1} = x_t - \alpha\nabla F(x_t)w_t$
7: **end for**

---

**Algorithm 2 warm-start**$(w_0, x_0, \rho)$

---
1: **for** $n = 0, 1, ..., N-1$ **do**
2:      $w_{n+1} = \Pi_{\mathcal{W}}\left(w_n - \beta'[\nabla F(x_0)^\top \nabla F(x_0)w_n + \rho w_n]\right)$
3: **end for**
4: **Output** $w_N$

---

### 3.2 STOCHASTIC MGDA WITH DOUBLE SAMPLING

In the stochastic setting, our algorithm keeps the same structure, having a warm start process if we aim to control the CA distance in every iteration. In Algorithm 2, we do the same projected gradient descent without using stochastic gradients. This is because we only need to compute $\nabla F(x_0)^\top \nabla F(x_0)$ once and reuse it in the whole loop, which does not bring a computational burden. Then in the update loop, we update the weight and model parameters accordingly. We use a double-sampling strategy here to make the weight gradient estimator unbiased (Xiao et al., 2024) such that $d_t$ is a near-unbiased multi-gradient $\mathbb{E}[\nabla G_2(x_t)^\top \nabla G_3(x_t)w_t + \rho w_t] = \nabla F(x_t)^\top \nabla F(x_t)w_t + \rho w_t$, where $\nabla G_2(x_t)$ and $\nabla G_3(x_t)$ are independent and unbiased estimates of $\nabla F(x_t)$. Similarly, we do not involve a warm start process if we require the average CA distance to be small.

### 3.3 MGDA WITH FAST APPROXIMATION

It can be seen from Algorithm 1 and Algorithm 3 that MGDA requires $\mathcal{O}(K)$ space and time to compute and store all task gradients at each iteration for updating the weight $w_t$. This becomes a drawback when the number of tasks or the model size is large. Motivated by Liu et al. (2024), one solution is to use the Taylor Theorem to approximate the gradient for updating the weight $w_t$ as

$$F(x_t) - F(x_{t+1}) = \nabla F(x_t)^\top(x_t - x_{t+1}) - R(x_t) = \alpha\nabla F(x_t)^\top \nabla F(x_t)w_t - R(x_t),$$

where $R(x_t)$ is the remainder term and it takes the order $R(x_t) = o(\|x_t - x_{t+1}\|^2)$, which can be made sufficiently small by adjusting the step size. By incorporating this fast approximation (FA)

in Algorithm 1, we then present MGDA-FA in Algorithm 4 (shown in Appendix C), where $x_t$ is updated along the update direction $d_t = \nabla F(x_t)w_t$ to get $x_{t+1}$ following by the update rule of $w_t$

$$w_{t+1} = \Pi_{\mathcal{W}}\left(w_t - \beta\left[\frac{F(x_t) - F(x_{t+1})}{\alpha} + \rho w_t\right]\right). \tag{6}$$

As a result, in the model parameters update process, MGDA-FA only requires one backward process by calculating the gradient of $F(x_t)w_t$ w.r.t. $x_t$ without storing it, and additional forward processes to compute $F(x_{t+1})$ in the weight update process. This approach saves computational and memory costs in the practical implementation significantly. **Most importantly, we provide a theoretical guarantee for this efficient method (in eq. (6))**.

---

**Algorithm 3** Stochastic MGDA with Double Sampling

1: **Initialize:** model parameters $x_0$, weights $w_0$ and a constant $\rho$
2: *Option I: $w_0$=**warm-start**$(w_0, x_0, \rho)$ # for analyzing iteration-wise CA distance*
3: *Option II: $w_0 \leftarrow w_0$*          # for analyzing averaged CA distance
4: **for** $t = 0, 1, ..., T-1$ **do**
5:     $x_{t+1} = x_t - \alpha\nabla G_1(x_t)w_t$
6:     $w_{t+1} = \Pi_{\mathcal{W}}\left(w_t - \beta[\nabla G_2(x_t)^\top \nabla G_3(x_t)w_t + \rho w_t]\right)$ # double sampling
7: **end for**

---

## 4 CONVERGENCE ANALYSIS UNDER AVERAGE CA DISTANCE

In this section, we provide the theoretical results for Algorithms 1 and 3 **without warm starts** to obtain an $\epsilon$-accurate Pareto stationary point, with the average CA distance over iterations in $\mathcal{O}(\epsilon)$.

### 4.1 DETERMINISTIC SETTING

**Assumption 1.** *Each objective function $f_i \ \forall i \in [K]$ is twice differentiable and lower bounded by $f_i^* := \inf_{x \in \mathbb{R}^m} f_i(x) > -\infty$.*

**Assumption 2.** *Each objective function $f_i \ \forall i \in [K]$ is generalized $\ell$-smooth defined in Definition 1, where $\ell : [0, +\infty) \to (0, +\infty)$ is a continuous non-decreasing function such that $\varphi(a) = \frac{a^2}{2\ell(2a)}$ is monotonically increasing for any $a \geq 0$.*

These assumptions are the most relaxed ones in existing MOO works since they directly assume objective smoothness or gradient/function value boundness (Liu et al., 2021; Fernando et al., 2022; Navon et al., 2022; Xiao et al., 2024; Chen et al., 2024; Yang et al., 2024; Ban & Ji, 2024). It also includes the widely studied standard $L$-smoothness (Nemirovskij & Yudin, 1983; Ghadimi & Lan, 2013; Ghadimi et al., 2016), $(L_0, L_1)$-smoothness (Zhang et al., 2019) as special cases. Moreover, for any $0 \leq \gamma \leq 2$ and $x \in \mathcal{X}$, our assumption even holds for function $f$ such that $\|\nabla^2 f(x)\| \leq L_0 + L_1\|\nabla f(x)\|^\gamma$, where $\gamma$ are limited to $[0, 1]$ in Chen et al. (2023).

Let $c > 0$ and $F > 0$ be some constants such that $\Delta + c \leq F$, where $\Delta = \max_{i \in [K]}\{f_i(x_0) - f_i^*\}$. Define $M = \sup\{z \geq 0|\varphi(z) \leq F\}$. We then have the following convergence rate for Algorithm 1:

**Theorem 1.** *Let Assumptions 1 and 2 hold. Set $\beta = \mathcal{O}(\frac{1}{M^2}), \alpha = \mathcal{O}(\frac{1}{M^2} + \frac{1}{M\ell(M+1)}), T = \max\left(\Theta\left(\frac{1}{\alpha\epsilon^2}\right), \Theta\left(\frac{1}{\beta\epsilon^2}\right)\right)$ and $\rho = \mathcal{O}(\epsilon^2)$. We then have that $\frac{1}{T}\sum_{t=0}^{T-1}\|\nabla F(x_t)w_t\|^2 \leq \epsilon^2$.*

The full version with detailed constants and detailed proof can be found in Appendix D.1. Theorem 1 provides the first convergence rate to obtain an $\epsilon$-accurate Pareto stationary point for MOO problems with generalized $\ell$-smooth objectives. Moreover, it achieves the optimal sample complexity in the order of $\mathcal{O}(\epsilon^{-2})$ for GD with a single standard $L$-smooth objective (Carmon et al., 2020). The MOO problems with generalized $\ell$-smooth objectives are challenging due to **two reasons**: 1) $\|\nabla F(x)\|$ is potentially unbounded in our generalized $\ell$-smoothness setting, making all existing analysis in MOO (Liu et al., 2021; Fernando et al., 2022; Navon et al., 2022; Xiao et al., 2024; Chen et al., 2024; Yang et al., 2024; Ban & Ji, 2024) not applicable. 2) the update of $x$ includes all gradient information from each task, making the existing adaptive methods for single generalized smooth functions invalid.

To solve the challenges in Theorem 1, we find that a bounded function value implies a bounded gradient norm. Thus in our proof, we use induction to show that with parameters selected in

Theorem 1, for any $w \in \mathcal{W}$ and $t \leq T$, we have that $F(x_t)w$ is upper bounded by $F$. Consequently, for any $i \in [K]$, we have that $\|\nabla f_i(x)\| \leq M$, which solves the unbounded gradient norm problem in our generalized smoothness setting. Then we can show that $\|\nabla F(x_t)w_t\|$ converges.

**Corollary 1.** *Under the same setting in Theorem 1, $\frac{1}{T}\sum_{t=0}^{T-1}\|\nabla F(x_t)w_t - \nabla F(x_t)w_t^*\|^2 = \mathcal{O}(\epsilon^2)$.*

The proof is available in Appendix D.2. Corollary 1 shows that the average CA distance converges.

### 4.2 STOCHASTIC SETTING

In the stochastic setting, we assume that we have access to an unbiased stochastic gradient $\nabla f_i(x; s)$ instead of the true gradient $\nabla f_i(x)$, where $s$ is the collected samples. To prove convergence, we have the following assumption.

**Assumption 3.** *There exists some $\sigma \geq 0$ such that $\mathbb{E}[\|\nabla f_i(x; s) - \nabla f_i(x)\|^2] \leq \sigma^2$ for any $i \in [K]$.*

Assumption 3 indicates bounded gradient variances, which is widely studied (Xiao et al., 2024; Li et al., 2024a; Fernando et al., 2022). At time $t$ in the stochastic model, let $s_{t,i} = (s_{t,i,1}, s_{t,i,2}, ..., s_{t,i,k})$ be the $i$-th collection of samples and $F(x_t; s_{t,i}) = (f_1(x_t; s_{t,i,1}), f_2(x_t; s_{t,i,2}), ..., f_k(x_t; s_{t,i,k}))$. Note that $i \in [3]$ because we have 3 samples in Algorithm 3. We choose $G_i(x_t) = F(x_t; s_{t,i})$. Define $\varepsilon_{t,i} = (\varepsilon_{t,i,1}, \varepsilon_{t,i,2}, ..., \varepsilon_{t,i,k}) = \nabla F(x_t) - \nabla G_i(x_t)$.

Let $F, c > 0$ and $0 < \delta \leq \frac{1}{2}$ be some constants such that $F \geq \frac{4(\Delta+c)}{\delta}$ and $M = \sup\{z \geq 0 | \varphi(z) \leq F\}$. Define the following random variables $\tau_1 = \min\{t | \exists i \in [K], f_i(x_{t+1}) - f_i^* > F\} \wedge T$, $\tau_2 = \min\{t | \exists i \in [K], j \in [3], \|\varepsilon_{t,j,i}\| > \frac{L_0}{\sqrt{\alpha\rho}}\} \wedge T$, $\tau_3 = \min\{t | \exists i, j \in [K], \|\varepsilon_{t,2,i}\|\|\varepsilon_{t,3,j}\| > \frac{L_1}{\sqrt{\alpha\rho}}\} \wedge T$ and $\tau = \min\{\tau_1, \tau_2, \tau_3\}$, where $L_0, L_1 > 0$ are some constants and $a \wedge b$ denotes $\min(a, b)$. We have the following theorem for Algorithm 3::

**Theorem 2.** *Let Assumptions 1, 2, 3 hold. Set $\rho = \mathcal{O}(\delta^2\epsilon^2), \beta = \mathcal{O}(\min\{\frac{\delta\epsilon^2}{M^4}, \frac{\rho}{M^2}\}), \alpha = \mathcal{O}(\min\{\beta, \frac{\rho}{\ell(M+1)^2}, \frac{\rho}{M^2}, \frac{\delta}{\rho T}, \frac{1}{\beta T M^2}\})$ and $T = \Theta(\max\{\frac{1}{\delta\alpha\epsilon^2}, \frac{M^2}{\delta^2\epsilon^4}\})$. We then have that with the probability at least $1 - \delta$, $\frac{1}{T}\sum_{t=0}^{T-1}\|\nabla F(x_t)w_t\|^2 = \mathcal{O}(\epsilon^2)$ .*

The full version and detailed proof can be found in Appendix D.3. When we set $\alpha, \beta, \rho = \mathcal{O}(\epsilon^2)$ and $T = \Theta(\epsilon^{-4})$, we can find an $\epsilon$-stationary point with the optimal sample complexity in the order of $\mathcal{O}(\epsilon^{-4})$ for SGD with a single $L$-smooth objective (Arjevani et al., 2023). Note that in the proof of Theorem 1, we show for each $t \leq T$ and $w \in W$, we have that $F(x_t)w$ is bounded by applying a small constant step size $\alpha$ and $\beta$. However, this condition does not necessarily hold for our stochastic setting due to the unbounded gradient noise. To solve this problem, we introduce stopping time $\tau$. The advantages are as follows: 1) for any $t \leq \tau$, $w \in W$, we have that $F(x_t)w$ is bounded; 2) for any $t < \tau$, the norm of gradient noise is bounded; 3) due to the optional stopping theorem, for any $w \in W$ and $i \in [3]$, we have that $\mathbb{E}[\sum_{t=0}^{\tau}\varepsilon_{t,i}w] = 0$. Thus, we can further get the following lemma:

**Lemma 1.** *Using the parameters selected in Theorem 2, we have that $\mathbb{E}[F(x_\tau)w] - F^*w \leq \frac{\delta F}{8} - \frac{\alpha}{2}\mathbb{E}\left[\sum_{t=0}^{\tau-1}\|\nabla F(x_t)w_t\|^2\right]$ .*

The proof of Lemma 1 is available in D.4. Lemma 1 indicates that $\frac{\alpha}{2}\mathbb{E}\left[\sum_{t=0}^{\tau-1}\|\nabla F(x_t)w_t\|^2\right]$ is bounded by some constant and if $\tau = T$ with high probability, we have that $\frac{1}{T}\mathbb{E}\left[\sum_{t=0}^{T-1}\|\nabla F(x_t)w_t\|^2\Big|\tau = T\right] = \mathcal{O}\left(\frac{1}{\alpha T}\right)$. Note that $\{\tau < T\} = \{\tau_2 < T\} \cup \{\tau_3 < T\} \cup \{\tau_1 < T, \tau_2 = T, \tau_3 = T\}$. The first two events are related to the gradient noise, where the probabilities can be bounded by Assumption 3 and Chebyshev's inequality. The last event indicates that for some $i \in [K]$, we have $f_i(x_\tau) - f_i^* \leq \frac{F}{2}$. Based on Lemma 1 and Markov inequality, we can show that $\mathbb{P}(\{\tau_1 < T, \tau_2 = T, \tau_3 = T\}) \leq \frac{\delta}{4}$ and we can further show that $\mathbb{P}(\tau = T) \geq 1 - \frac{\delta}{2}$. We then have that $\frac{1}{T}\sum_{t=0}^{T-1}\|\nabla F(x_t)w_t\|^2$ converges with high probability. Similar to Corollary 1, Theorem 2 also implies the average of CA distances converges with time with high probability.

## 5 CONVERGENCE ANALYSIS UNDER ITERATION-WISE CA DISTANCE

In Section 4 we show that the average CA distance is bounded under generalized smooth conditions. The average CA distance is also studied in MoCo (Fernando et al., 2022) and MoDo (Chen et al.,

2024) with the bounded gradient assumption and these works only focus on guarantees of the average CA distance over iterations. However, a $\epsilon$-level average CA distance only implies the smallest CA distance to be $\epsilon$-level. Since we want to keep the update direction close enough to the CA direction, it is better to have a tighter bound of CA distances. In this section, we show the iterative CA distance is $\mathcal{O}(\epsilon)$ **with a warm-start process** and convergence results for Algorithms 1, 3, and 4.

## 5.1 DETERMINISTIC SETTING

**Deterministic setting without fast approximation.** We first provide results about bounded iteration-wise CA distance for Algorithm 1 with a warm start.

**Theorem 3.** *Let Assumptions 1 and 2 hold. Set $\beta' \leq \frac{1}{M^2}, \rho = \mathcal{O}(\epsilon^2), \beta = \mathcal{O}(\epsilon^4), \alpha = \mathcal{O}(\epsilon^9),$ $N = \Omega(\epsilon^{-2})$ as constants, and $T = \Theta(\epsilon^{-11})$. All the parameters satisfy the requirements in the formal version of Theorem 1 and we have $\|\nabla F(x_t)w_t - \nabla F(x_t)w_t^*\| = \mathcal{O}(\epsilon)$.*

The finite time error bound and the full proof can be found in Appendix E.1. Since our parameters satisfy the requirements in the formal version of Theorem 1, we can find an $\epsilon$-accurate Pareto stationary point with $\mathcal{O}(\epsilon^{-11})$ samples. In the analysis of CA distance, we show that the CA distance can be bounded by the term $\|w_t - w_{t,\rho}^*\|$ plus the strongly-convex constant $\rho$. Meanwhile, there is a decay relation between $\|w_{t+1} - w_{t+1,\rho}^*\|$ and $\|w_t - w_{t,\rho}^*\|$ with some error terms controlled by step sizes. Nevertheless, the error terms will accumulate since we do telescoping on this decay relation, which will be the dominating term. Thus, step sizes have to be much smaller than the choices in Theorem 1 to guarantee iteration-wise small CA distance.

**Deterministic setting with fast approximation.** In this section, we show the convergence rate of Algorithm 4 and bounded iteration-wise CA distance.

**Theorem 4.** *Let Assumptions 1 and 2 hold. Set $N = \Omega(\epsilon^{-2})$, $\beta' \leq \frac{1}{M^2}, \rho = \mathcal{O}\left(\min\left\{\epsilon^2, \frac{1}{\alpha T}\right\}\right), \beta = \mathcal{O}(\epsilon^2), \alpha = \mathcal{O}\left(\min\{\beta, \epsilon^2, \frac{1}{\beta T}\}\right)$ as constants, $T = \Theta\left(\max\{\frac{1}{\alpha\epsilon^2}, \frac{1}{\beta\epsilon^2}\}\right)$. We have $\frac{1}{T}\sum_{t=0}^{T-1}\|\nabla F(x_t)w_t\|^2 = \mathcal{O}(\epsilon^2)$.*

The full version and proof can be found in Appendix E.2. We can easily extend the analysis in Appendix D.1 to the convergence analysis of Algorithm 4 under the average CA distance, because the only extra effort is dealing with the remainder term, which can be bounded by the smallest step size. As a result, the sample complexity remains the same $\mathcal{O}(\epsilon^{-11})$ to achieve a Pareto stationary point.

**Theorem 5.** *Let Assumptions 1 and 2 hold. We choose $\beta' \leq \frac{1}{M^2}, \rho = \mathcal{O}(\epsilon^2), \beta = \mathcal{O}(\epsilon^4), \alpha = \mathcal{O}(\epsilon^9),$ $N = \Omega(\epsilon^{-2})$ as constants, and $T = \Theta(\epsilon^{-11})$. We have that $\|\nabla F(x_t)w_t - \nabla F(x_t)w_t^*\| = \mathcal{O}(\epsilon)$.*

## 5.2 STOCHASTIC SETTING

In this section, we show that Algorithm 3 with a warm start and mini-batches achieves a bounded iteration-wise CA distance with high probability. In this section, we choose $G_i(x_t) = \frac{1}{n_s}\sum_{i=n_s i-n_s+1}^{n_s i} F(x_t; s_{t,i})$, where $n_s$ is the size of the mini-batch.

**Theorem 6.** *Let Assumptions 1, 2 and 3 hold. Set $\beta' \leq \frac{1}{M^2}, \alpha = \mathcal{O}(\epsilon^9), \beta = \mathcal{O}(\epsilon^4), \rho = \mathcal{O}(\epsilon^2),$ $n_s = \Omega(\epsilon^{-6}), N = \Omega(\epsilon^{-2}),$ and $T = \Theta(\epsilon^{-11})$, and all the parameters satisfy the requirements in Theorem 2. We then have $\|\nabla F(x_t)w_t - \nabla F(x_t)w_t^*\| = \mathcal{O}(\epsilon)$, with the probability at least $1 - \delta$.*

The full version with detailed constants and proof can be found in Appendix E.4. Since our parameters satisfy all requirements in Theorem 2, we can find an $\epsilon$-accurate Pareto stationary point with high probability. Compared with Theorem 2, to guarantee an iteration-wise CA distance, despite our warm start process, a mini-batch method is required in our analysis. This is because given $\tau = T$, the gradient is not unbiased. In Theorem 2, the optional stopping theorem is applied which indicates that the expectation of the cumulative gradient is zero. However, for each iteration, this optional stopping theorem does not hold and the estimated error is controlled by the size of the mini-batch. Then, the sample complexity to get a Pareto stationary point becomes $\mathcal{O}(\epsilon^{-17})$ due to necessary mini-batch $n_s$.

## 6 EXPERIMENTS

In this experiment, we evaluate the performance of the Cityscapes(Cordts et al., 2016) and NYU-v2 (Silberman et al., 2012) datasets. The former involves 2 pixel-wise tasks: 7-class semantic

| Method | Segmentation | | Depth | | Surface Normal | | | | | MR ↓ | Δm% ↓ |
|---|---|---|---|---|---|---|---|---|---|---|---|
| | | | | | Angle Distance ↓ | | Within $t°$ ↑ | | | | |
| | mIoU ↑ | Pix Acc ↑ | Abs Err ↓ | Rel Err ↓ | Mean | Median | 11.25 | 22.5 | 30 | | |
| STL | 38.30 | 63.76 | 0.6754 | 0.2780 | 25.01 | 19.21 | 30.14 | 57.20 | 69.15 | | |
| LS | 39.29 | 65.33 | 0.5493 | 0.2263 | 28.15 | 23.96 | 22.09 | 47.50 | 61.08 | 7.89 | 5.59 |
| SI | 38.45 | 64.27 | 0.5354 | 0.2201 | 27.60 | 23.37 | 22.53 | 48.57 | 62.32 | 7.33 | 4.39 |
| RLW (Lin et al., 2021) | 37.17 | 63.77 | 0.5759 | 0.2410 | 28.27 | 24.18 | 22.26 | 47.05 | 60.62 | 9.89 | 7.78 |
| DWA (Liu et al., 2019) | 39.11 | 65.31 | 0.5510 | 0.2285 | 27.61 | 23.18 | 24.17 | 50.18 | 62.39 | 7.11 | 3.57 |
| UW (Kendall et al., 2018) | 36.87 | 63.17 | 0.5446 | 0.2260 | 27.04 | 22.61 | 23.54 | 49.05 | 63.65 | 7.11 | 4.05 |
| MGDA (Désidéri, 2012) | 30.47 | 59.90 | 0.6070 | 0.2555 | **24.88** | **19.45** | 29.18 | **56.88** | **69.36** | 5.56 | 1.38 |
| MoCo (Fernando et al., 2022) | 40.30 | 66.07 | 0.5575 | **0.2135** | 26.67 | 21.83 | 25.61 | 51.78 | 64.85 | 5.00 | 0.16 |
| MoDo (Chen et al., 2024) | 35.28 | 62.62 | 0.5821 | 0.2405 | 25.65 | 20.33 | 28.04 | 54.86 | 67.37 | 7.55 | 0.49 |
| Nash-MTL Navon et al. (2022) | 40.13 | 65.93 | 0.5261 | 0.2171 | 25.26 | 20.08 | 28.40 | 55.47 | 68.15 | 3.33 | -4.04 |
| FAMO (Liu et al., 2024) | 38.88 | 64.90 | 0.5474 | 0.2194 | 25.06 | 19.57 | **29.21** | 56.61 | 68.98 | 3.22 | -4.10 |
| MGDA-warm start | **40.57** | **67.17** | **0.5240** | 0.2281 | 25.21 | 19.74 | 28.74 | 55.79 | 68.21 | **2.78** | **-4.42** |

Table 2: Multi-task supervised learning on NYU-v2 dataset for different MOO methods comparison.

segmentation (Task 1) and depth estimation (Task 2) while the latter involves 3 pixel-wise tasks: 13-class semantic segmentation, depth estimation and surface normal estimation. Following the same experiment setup of (Xiao et al., 2024), we build a SegNet (Badrinarayanan et al., 2017) as the model. We compare the performance of MGDA **with a warm start**, which is used to ensure a small CA distance per iteration and mitigate gradient conflicts, against **popular MGDA-type methods** including MGDA (Désidéri, 2012), PCGrad (Yu et al., 2020), GradDrop (Chen et al., 2020), CAGrad (Liu et al., 2021), MoCo (Fernando et al., 2022), MoDo (Chen et al., 2024), Nash-MTL (Navon et al., 2022), and FAMO (Liu et al., 2024). *Since SDMGrad (Xiao et al., 2024) incorporates a preference-based regularization within the MGDA framework, we exclude it from the tables to ensure a fair comparison.* We utilize the metric $\Delta m\%$ to reflect the overall performance, which considers the average per-task performance drop versus the single-task (STL) baseline to assess methods. It can be observed in Table 2 and Table 3 that MGDA-warm start has a much more balanced performance. Meanwhile, the proposed MGDA-FA is much faster as shown in Table 5 in the Appendix.

We also illustrate the relationship between the gradient norm and the local smoothness for each task of the Cityscapes dataset. To do this, we compute them according to the method provided in Section H.3 in Zhang et al. (2019). We scatter the local smoothness constant against gradient norms in Figure 1 for the semantic segmentation task and depth estimation task in Figure 2 (in the appendix), respectively. Both results demonstrate a positive correlation between them, which further substantiates the necessity of our analysis. More experimental details can be found in Appendix B.

| Method | Segmentation | | Depth | | Δm% ↓ |
|---|---|---|---|---|---|
| | mIoU ↑ | Pix Acc ↑ | Abs Err ↓ | Rel Err ↓ | |
| STL | 74.01 | 93.16 | 0.0125 | 27.77 | |
| MGDA (Désidéri, 2012) | 68.84 | 91.54 | 0.0309 | 33.50 | 44.14 |
| PCGrad (Yu et al., 2020) | 75.13 | 93.48 | 0.0154 | 42.07 | 18.29 |
| GradDrop (Chen et al., 2020) | 75.27 | 93.53 | 0.0157 | 47.54 | 23.73 |
| CAGrad (Liu et al., 2021) | 75.16 | 93.48 | 0.0141 | 37.60 | 11.64 |
| MoCo (Fernando et al., 2022) | **75.42** | 93.55 | 0.0149 | 34.19 | 9.90 |
| MoDo (Chen et al., 2024) | 74.55 | 93.32 | 0.0159 | 41.51 | 18.89 |
| Nash-MTL (Navon et al., 2022) | 75.41 | **93.66** | **0.0129** | 35.02 | 6.82 |
| FAMO (Liu et al., 2024) | 74.54 | 93.29 | 0.0145 | 32.59 | 8.13 |
| MGDA-warm start | 75.41 | 93.46 | 0.0133 | **31.07** | **3.93±1.19** |

Table 3: Multi-task learning on Cityscapes dataset.

## 7 CONCLUSION

Building upon our observations of MGDA convergence under the $(L_0, L_1)$ smoothness condition, this paper provides a rigorous convergence analysis for both the fundamental MGDA and its stochastic variant under a more challenging, relaxed, and practical generalized $\ell$-smoothness assumption. Furthermore, we introduce a warm start progress to provide a more precise control over the iteration-wise CA distance. Our analysis also shows that an efficient variant named MGDA-FA, which uses only $\mathcal{O}(1)$ time and space, achieves the same performance guarantee as MGDA. We anticipate that the convergence analysis developed in this work will provide valuable insights for analyzing other MOO algorithms such as CAGrad, PCGrad, FairGrad and FAMO under the generalized smoothness condition. The warm-start strategy may be of independent interest to other single-loop MOO algorithms to achieve a sufficiently small iteration-wise CA distance.

## ACKNOWLEDGMENTS

The work of Q. Zhang and S. Zou was partially supported by NSF under Grant CCF-2438429. P. Xiao and K. Ji were partially supported by NSF grants CCF-2311274 and ECCS-2326592.

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

## A  NOTATION SUMMARY

We have summarized the notations used in this paper in the following table.

Table 4: Notations and their descriptions.

| Notations | Descriptions |
|---|---|
| $x \in \mathbb{R}^m$ | Model parameter, or decision variable |
| $s_{t,i,k}$ | i-th collection of samples at time t in the stochastic model for training or testing |
| $f_k(x)$ | A scalar-valued population objective function |
| $\nabla f_k(x)$ | Gradient of $f_k(x)$, with $\nabla f_k(x) : \mathbb{R}^m \mapsto \mathbb{R}^m$ |
| $F(x)$ | A vector-valued population objective function |
| $\nabla F(x)$ | Gradient of $F(x)$, with $\nabla F(x) : \mathbb{R}^d \mapsto \mathbb{R}^{d \times K}$ |
| $\ell(a)$ | A continuous non-decreasing function: $[0, +\infty) \to (0, +\infty)$ |
| $\varphi(a)$ | A monotonically increasing for any $a \geq 0$: $\varphi(a) = \frac{a^2}{2\ell(2a)}$ |
| $w \in \mathcal{W}$ | Weighting parameter in a probability simplex over $[K]$, |
| $w_\rho^* \in \mathcal{W}$ | optimal solution to equation 5, when $\rho = 0$, it is simplified as $w^*$ |
| $\alpha$ | Step size to update model parameter $x$ |
| $\beta$ | Step size to update weight in the main loop |
| $\beta'$ | Step size to update weight in the warm start |
| $\rho$ | Regularization parameter in equation 5 |

## B  EXPERIMENTAL DETAILS

### B.1  RELATION BETWEEN GRADIENT NORMS AND THE LOCAL SMOOTHNESS

We show the relation between local smoothness and gradient norms of each task in this part. Both results demonstrate a positive correlation between them, which further substantiates the necessity of our analysis.

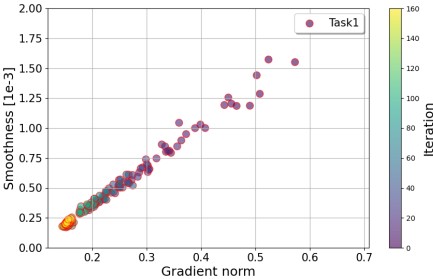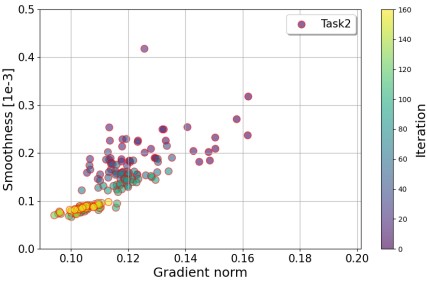

Figure 2: Local smoothness constant vs. Gradient norm on training SegNet on CityScapes dataset of each task. Task 1 on the left and Task 2 on the right.

### B.2  RESULTS AND RUNNING TIME COMPARISON BETWEEN MGDA-WARM START AND MGDA-FA

We compare the results and average running time of the proposed algorithms, MGDA-warm start and MGDA-FA of the Cityscapes (Cordts et al., 2016). The time in Table 5 is an average of the total running time over epochs (in minutes). The result solidifies the advantage of the fast approximation. For its performance, the fast approximation may introduce substantial errors in practice, potentially leading to inaccuracies in the weight update process. Meanwhile, the convergence analysis of the stochastic variant is not revealed. However, to enhance its effectiveness, we could apply a logarithmic technique in FAMO(Liu et al., 2024).

| Method | Segmentation | | Depth | | $\Delta m\% \downarrow$ | Average running time |
|---|---|---|---|---|---|---|
| | mIoU $\uparrow$ | Pix Acc $\uparrow$ | Abs Err $\downarrow$ | Rel Err $\downarrow$ | | |
| STL | 74.01 | 93.16 | 0.0125 | 27.77 | | |
| MGDA-warm start | 75.41 | 93.46 | 0.0133 | 31.07 | 3.93 | 2.93 |
| MGDA-FA | 74.38 | 93.24 | 0.0160 | 41.78 | 19.44 | 1.93 |

Table 5: Multi-task learning on Cityscapes dataset.

### B.3 RESULTS OF OTHER MOO METHODS WITH WARM START

The warm start strategy can be applied to other multi-objective optimization algorithms. For MGDA-based methods, both MoCo (Fernando et al., 2022) and MoDo (Chen et al., 2024) can incorporate the warm start as an add-on. However, the MoDo-warm start will exactly recover to our algorithm. Therefore, we evaluate the performance of the MoCo-warm start on the Cityscapes (Cordts et al., 2016) and NYU-v2 (Silberman et al., 2012) datasets. As shown in the following tables, the warm start strategy improves performance.

| Method | Segmentation | | Depth | | $\Delta m\% \downarrow$ |
|---|---|---|---|---|---|
| | mIoU $\uparrow$ | Pix Acc $\uparrow$ | Abs Err $\downarrow$ | Rel Err $\downarrow$ | |
| MoCo | 75.42 | 93.55 | 0.0149 | 34.19 | 9.90 |
| MoCo-warm start | 75.48 | 93.54 | 0.0148 | 31.43 | 7.32 |

Table 6: Multi-task supervised learning with MoCo-warm start on Cityscapes dataset.

| Method | Segmentation | | Depth | | Surface Normal | | | | | $\Delta m\% \downarrow$ |
|---|---|---|---|---|---|---|---|---|---|---|
| | mIoU $\uparrow$ | Pix Acc $\uparrow$ | Abs Err $\downarrow$ | Rel Err $\downarrow$ | Angle Distance $\downarrow$ | | Within $t° \uparrow$ | | | |
| | | | | | Mean | Median | 11.25 | 22.5 | 30 | |
| MoCo | 40.30 | 66.07 | 0.5575 | 0.2135 | 26.67 | 21.83 | 25.61 | 51.78 | 64.85 | 0.16 |
| MoCo-warm start | 38.40 | 64.40 | 0.5377 | 0.2315 | 26.04 | 20.57 | 27.11 | 54.02 | 66.63 | -0.88 |

Table 7: Multi-task supervised learning with MoCo-warm start on NYU-v2 dataset.

### B.4 IMPLEMENTATION DETAILS

**Multi-task learning on Cityscapes dataset.** Following the experiment setup in Xiao et al. (2024), we train our method for 200 epochs, using SGD optimizers for both model parameters and weights, and the batch size for Cityscapes is 8. We compute the averaged test performance over the last 10 epochs as the final performance measure. We fix the $\beta = 0.5$ and do a grid search on hyperparameters including $N \in [10, 20, 40, 50], \alpha \in [0.0001, 0.0002, 0.0005, 0.001]$, and $\rho \in [0.01, 0.05, 0.1, 0.2, 0.5, 0.6, 0.7, 0.8, 0.9, 1]$ and choose the best result from them. It turns out our best performance is based on the choice that $N = 40, \alpha = 0.0005, \beta = 0.5$, and $\rho = 0.5$. The choice of hyperparameters for MGDA-FA turns out to be the same as that for MGDA-warm start. All experiments are run on NVIDIA RTX A6000.

**Multi-task learning on NYU-v2 dataset.** Following the experiment setup in Xiao et al. (2024), we train our method for 200 epochs, using SGD optimizers for both model parameters and weights, and the batch size for NYU-v2 is 2. We compute the averaged test performance over the last 10 epochs as the final performance measure. We fix the $\beta = 0.5$ and do a grid search on hyperparameters including $N \in [10, 20, 40, 50], \alpha \in [0.0001, 0.0002, 0.0005, 0.001]$, and $\rho \in [0.01, 0.05, 0.1, 0.2, 0.5, 0.6, 0.7, 0.8, 0.9, 1]$ and choose the best result from them. It turns out our best performance is based on the choice that $N = 40, \alpha = 0.0001, \beta = 0.5$, and $\rho = 0.5$. All experiments are run on NVIDIA RTX A6000.

$\Delta m\%$ reflects the average per-task performance drop versus the single-task (STL) baseline $b$ to assess method $m$. We calculate it by the following equation

$$\Delta m\% = \frac{1}{K} \sum_{k=1}^{K} (-1)^{l_k} (M_{m,k} - M_{b,k})/M_{b,k} \times 100,$$

where $K$ is the number of metrics, $M_{b,k}$ is the value of metric $M_k$ obtained by baseline $b$, and $M_{m,k}$ obtained by the compared method $m$. $l_k = 1$ if the evaluation metric $M_k$ on task $k$ prefers a higher value and $0$ otherwise.

**Generalized smoothness illustration.** To illustrate the relation between gradient norms and local smoothness, we follow the method in Section H.3 in Zhang et al. (2019)and run SGD on each task separately without the warm start process. The code is available in https://github.com/JingzhaoZhang/why-clipping-accelerates. Since there is no weight update process, we only need to choose $\alpha = 0.0005$ for both tasks.

## C  ALGORITHM

---
**Algorithm 4** MGDA with Fast Approximation (MGDA-FA)

---
1: **Initialize:** model parameters $x_0$, weights $w_0$ and a constant $\rho$
2: $w_0$=**Warm-start**$(w_0, x_0, \rho)$
3: **for** $t = 0, 1, ..., T - 1$ **do**
4:     $x_{t+1} = x_t - \alpha \nabla F(x_t) w_t$
5:     Update $w_t$ according to eq. (6)
6: **end for**

---

## D  DETAILED PROOFS FOR AVERAGE CA DISTANCE

### D.1  FORMAL VERSION AND PROOF OF THEOREM 1

Let $F > 0$ be some constants such that

$$\Delta + 3 \leq F. \tag{7}$$

Define $M = \sup\{z \geq 0 | \varphi(z) \leq F\}$. We then have the following convergence rate for Algorithm 1 without warm start:

**Theorem 7.** *Suppose Assumptions 1 and 2 are satisfied, and we choose constant step sizes that* $\beta \leq \frac{1}{4KM^2}, \alpha \leq \min\left\{\beta, \frac{1}{2\ell(M+1)}, \frac{1}{M\ell(M+1)}\right\}, T \geq \max\left\{(\frac{10\Delta}{\alpha\epsilon^2}, \frac{10}{\epsilon^2\beta}\}\right) = \Theta(\epsilon^{-2})$, *and* $\rho \leq \min\left(\frac{\epsilon^2}{20}, \sqrt{\frac{\epsilon^2}{10\beta}}, \frac{1}{2T\alpha}, \sqrt{\frac{1}{T\alpha\beta}}\right) = \mathcal{O}(\epsilon^2)$. *We have that*

$$\frac{1}{T} \sum_{t=0}^{T-1} \|\nabla F(x_t) w_t\|^2 \leq \epsilon^2. \tag{8}$$

*Proof.* Compared with the standard $L$-smoothness, the generalized smoothness is more challenging to address due to the unbounded Lipschitz constant. Lemma 2 demonstrates that a bounded function value implies a bounded gradient norm, which further implies a bounded Lipschitz constant. In the following, we solve the unbounded Lipschitz constant problem by showing that the function value is bounded with the parameters selected in Theorem 1. We prove that for any $i \in K$ and $t \leq T$ we have that $f_i(x_t) - f_i^* \leq F$ by induction.

**Base Case:** since $M$ is non-negative, according to equation 7 we have that $f_i(x_0) - f_i^* \leq \Delta \leq F$ holds for any $i \in [K]$.

**Induction step:** assume that for any $i \in [K]$ and $t \leq k < T$, we have that $f_i(x_t) - f_i^* \leq F$ holds. We then prove that $f_i(x_{k+1}) - f_i^* \leq F$ holds for any $i \in [K]$.

For $f_i(x_t) - f_i^* \leq F$, based on the monotonicity shown in Lemma 2, we have that $\|\nabla f_i(x_t)\| \leq M$. From assumption 2, we have that $f_i(x)$ is $\left(\frac{1}{\ell(\|\nabla f_i(x))\|+1)}, \ell(\|\nabla f_i(x)\|+1)\right)$-smooth by setting $a = 1$. For any $t \leq k$, we have that

$$\|x_{t+1} - x_t\| = \alpha\|\nabla F(x_t)w_t\| \leq \alpha M \leq \frac{1}{\ell(M+1)} \leq \frac{1}{\ell(\|\nabla f_i(x_t)\|+1)},$$

where the second inequality is due to $\alpha \leq \frac{1}{M\ell(M+1)}$ and the last inequality is due to $\|\nabla f_i(x_t)\| \leq M$. Based on Assumption 2, Definition 2 and Lemma 3.3 in (Li et al., 2024a), we have the following descent lemma:

$$f_i(x_{t+1}) \leq f_i(x_t) - \alpha\langle\nabla f_i(x_t), \nabla F(x_t)w_t\rangle + \frac{\ell(\|\nabla f_i(x_t)\|+1)}{2}\alpha^2\|\nabla F(x_t)w_t\|^2$$

$$\leq f_i(x_t) - \alpha\langle\nabla f_i(x_t), \nabla F(x_t)w_t\rangle + \frac{\ell(M+1)}{2}\alpha^2\|\nabla F(x_t)w_t\|^2.$$

As a result, for any $w \in \mathcal{W}$, we have that

$$F(x_{t+1})w \leq F(x_t)w - \alpha\langle\nabla F(x_t)w, \nabla F(x_t)w_t\rangle + \frac{\ell(M+1)}{2}\alpha^2\|\nabla F(x_t)w_t\|^2. \tag{9}$$

Based on the update process of $w$, we have that

$$w_{t+1} = \Pi_{\mathcal{W}}\Big(w_t - \beta\big(\nabla F(x_t)^\top\nabla F(x_t)w_t + \rho w_t\big)\Big).$$

It then follows that

$$\|w_{t+1} - w\|^2$$
$$= \left\|\Pi_{\mathcal{W}}\Big(w_t - \beta\big(\nabla F(x_t)^\top\nabla F(x_t)w_t + \rho w_t\big)\Big) - w\right\|^2$$
$$\leq \left\|\Big(w_t - \beta\big(\nabla F(x_t)^\top\nabla F(x_t)w_t + \rho w_t\big)\Big) - w\right\|^2$$
$$= \|w_t - w\|^2 - 2\beta\left\langle w_t - w, (\nabla F(x_t)^\top\nabla F(x_t) + \rho I)w_t\right\rangle$$
$$+ \beta^2\left\|(\nabla F(x_t)^\top\nabla F(x_t) + \rho I)w_t\right\|^2,$$

where the inequality is due to the non-expansiveness of projection. By rearranging the above inequality, we have that

$$\langle w_t - w, \nabla F(x_t)^\top\nabla F(x_t)w_t\rangle$$
$$\leq \frac{1}{2\beta}\left(\|w_t - w\|^2 - \|w_{t+1} - w\|^2\right) + 2\rho + \beta KM^2\|\nabla F(x_t)w_t\|^2 + \beta\rho^2. \tag{10}$$

Plug equation 10 into equation 9, and we can show that

$$F(x_{t+1})w - F(x_t)w$$
$$\leq -\alpha\|\nabla F(x_t)w_t\|^2 + \frac{\ell(M+1)}{2}\alpha^2\|\nabla F(x_t)w_t\|^2$$
$$+ \frac{\alpha}{2\beta}\left(\|w_t - w\|^2 - \|w_{t+1} - w\|^2\right) + \alpha\beta KM^2\|\nabla F(x_t)w_t\|^2 + \alpha\beta\rho^2 + 2\alpha\rho. \tag{11}$$

Taking sums of equation 11 from $t = 0$ to $k$, for any $w \in \mathcal{W}$ we have that

$$F(x_{k+1})w - F(x_0)w$$
$$\leq -\sum_{t=0}^{k}\alpha\|\nabla F(x_t)w_t\|^2 + \sum_{t=0}^{k}\left(\frac{\ell(M+1)}{2}\alpha^2 + \alpha\beta KM^2\right)\|\nabla F(x_t)w_t\|^2$$
$$+ \frac{\alpha}{2\beta}\|w_0 - w\|^2 + T\alpha\beta\rho^2 + 2T\alpha\rho$$
$$\leq \frac{\alpha}{2\beta}\|w_0 - w\|^2 + T\alpha\beta\rho^2 + 2T\alpha\rho, \tag{12}$$

where the first inequality is due to $k < T$ and the last inequality is due to that $\alpha \leq \frac{1}{2\ell(M+1)}$ and $\beta K M^2 \leq \frac{1}{4}$. Thus for any $i \in [K]$ it can be shown that

$$f_i(x_{t+1}) - f_i^* \leq f_i(x_0) - f_i^* + \frac{\alpha}{\beta} + T\alpha\beta\rho^2 + 2T\alpha\rho \leq F,$$

since we have that $\frac{\alpha}{\beta} \leq 1, T\alpha\beta\rho^2 \leq 1$ and $2T\alpha\rho \leq 1$. Now we finish the induction step and can show that $f_i(x_{k+1}) - f_i^* \leq F$ and equation 12 hold for all $k < T$ and $i \in [K]$.

Specifically, for $\alpha < \frac{1}{2\ell(M+1)}, \beta \leq \frac{1}{4KM^2}$, according to equation 12, for $k = T - 1$ we have that

$$\frac{1}{T}\sum_{t=0}^{T-1}\|\nabla F(x_t)w_t\|^2 \leq \frac{2(F(x_0)w - F^*w)}{\alpha T} + \frac{2}{\beta T} + 2\beta\rho^2 + 4\rho = \mathcal{O}(\epsilon^2),$$

which completes our proof. $\qquad\square$

**Lemma 2.** *(Lemma 3.5 in Li et al. (2024a)) If a function $f$ is $\ell$-smooth, we have that*

$$\varphi(\|\nabla f(x)\|) = \frac{\|\nabla f(x)\|^2}{2\ell(2\|\nabla f(x)\|)} \leq f(x) - f^*. \tag{13}$$

## D.2 Proof of Corollary 1

*Proof.* Recall that

$$\begin{aligned}
&\|\nabla F(x_t)w_t - \nabla F(x_t)w_t^*\|^2 \\
=&\|\nabla F(x_t)w_t\|^2 + \|\nabla F(x_t)w_t^*\|^2 - 2\langle\nabla F(x_t)w_t, \nabla F(x_t)w_t^*\rangle \\
\leq&\|\nabla F(x_t)w_t\|^2 + \|\nabla F(x_t)w_t^*\|^2 - 2\|\nabla F(x_t)w_t^*\|^2 \\
=&\|\nabla F(x_t)w_t\|^2 - \|\nabla F(x_t)w_t^*\|^2 \\
\leq&\|\nabla F(x_t)w_t\|^2,
\end{aligned}$$

where the first inequality holds because of the optimality condition of $w_t^* = \arg\min_{w\in\mathcal{W}}\frac{1}{2}\|\nabla F(x_t)w\|^2$ that for any $w \in \mathcal{W}$,

$$\langle w, \nabla F(x_t)^\top\nabla F(x_t)w_t^*\rangle \geq \langle w_t^*, \nabla F(x_t)^\top\nabla F(x_t)w_t^*\rangle = \|\nabla F(x_t)w_t^*\|^2.$$

Then we have

$$\frac{1}{T}\sum_{t=0}^{T-1}\|\nabla F(x_t)w_t - \nabla F(x_t)w_t^*\|^2 \leq \frac{1}{T}\sum_{t=0}^{T-1}\|\nabla F(x_t)w_t\|^2 = \mathcal{O}(\epsilon^2),$$

where we follow the same setting in Theorem 1. The proof is complete. $\qquad\square$

## D.3 Formal Version and Its Proof of Theorem 2

Let $c_1 > 0, c_2 > 0, c_3 > 0, c_4 > 0, c_5 > 0, c_6 > 0$ be some constants. Let $F > 0$ and $0 < \delta \leq \frac{1}{2}$ be some constants such that

$$F \geq \frac{8(\Delta + c_1 + c_2 + c_3 + c_4 + c_5 + c_6)}{\delta},$$

where $\Delta = \max_{i\in[d]}\{f_i(x_0) - f^*\}$. Let $L_0 > 0, L_1 > 0, b_1 > 0, b_2 > 0, b_3 > 0$ be some constants, for $B_1 = \min\left\{\frac{1}{4L_0^2\ell(M+1)^2}, \frac{b_1^2}{(3M\sqrt{K}L_0+M\sqrt{K}L_1)^2}, \frac{b_2}{\ell(M+1)L_0^2}\right\}$.

$\rho \leq \min\{\frac{\delta\epsilon^2}{48}, \sqrt{\frac{\delta\epsilon^2}{48}}, \frac{L_0^2\delta^2\epsilon^2}{1152K\sigma^2(\Delta+c_1)}, \frac{L_1^2\delta^2\epsilon^2}{384K\sigma^2(\Delta+c_1)}\} = \mathcal{O}(\delta^2\epsilon^2),$

$\beta \leq \min\{\frac{\delta\epsilon^2}{192K(M^2+\sigma^2)^2}, \frac{b_3\rho}{8KM^2L_0^2+4KL_1^2}, 1\} = \mathcal{O}(\min\{\frac{\delta\epsilon^2}{M^4}, \frac{\rho}{M^2}\}),$

$\alpha \leq \min\{\frac{\ell(M+1)}{2\max(1,M)}, c_1\beta, \rho B_1, \frac{c_2}{\sqrt{2T}M(3\sigma+\sigma^2)}, \frac{\sqrt{c_3}}{\sqrt{\ell(M+1)\sigma^2 T}}, \frac{\delta\epsilon^2}{48\sigma^2\ell(M+1)},$

$\frac{1}{\rho T} \min\{c_5, \frac{\delta L_0^2}{24K\sigma^2}, \frac{\delta L_1^2}{8K\sigma^4}\}, \frac{c_4}{4K\beta T(M^2+\sigma^2)^2}, \frac{c_6}{\rho\beta T}\} = \mathcal{O}(\min\{\beta, \frac{\rho}{\ell(M+1)^2}, \frac{\rho}{M^2}, \frac{\delta}{\rho T}, \frac{1}{\beta T M^2}\})$

and $T \geq \max\{\frac{48\Delta+48c_1}{\delta\alpha\epsilon^2}, \frac{4608M^2(3\sigma+\sigma^2)^2}{\delta^2\epsilon^4}\} = \mathcal{O}(\max\{\frac{1}{\delta\alpha\epsilon^2}, \frac{M^2}{\delta^2\epsilon^4}\})$ such that

$$b_1 + b_2 + b_3 + c_1 + \alpha\rho(1+\beta\rho) + 4\alpha\beta KM^4 \leq \frac{F}{2}. \tag{14}$$

Define the following random variables

$$\tau_1 = \min\{t | \exists i \in [K], f_i(x_{t+1}) - f_i^* > F\} \wedge T,$$

$$\tau_2 = \min\{t | \exists i \in [K], j \in [3], \|\varepsilon_{t,j,i}\| > \frac{L_0}{\sqrt{\alpha\rho}}\} \wedge T,$$

$$\tau_3 = \min\{t | \exists i, j \in [K], \|\varepsilon_{t,2,i}\|\|\varepsilon_{t,3,j}\| > \frac{L_1}{\sqrt{\alpha\rho}}\} \wedge T,$$

$$\tau = \min\{\tau_1, \tau_2, \tau_3\}.$$

We then have the following theorem:

**Theorem 8.** *If Assumptions 1, 2 and 3 hold, with the parameters selected above, we have that*

$$\frac{1}{T} \sum_{t=0}^{T-1} \|\nabla F(x_t)w_t\|^2 = \mathcal{O}(\epsilon^2),$$

*with the probability at least* $1 - \delta$.

*Proof.* **Small probability of the event** $\{\tau < T\}$.

We first show that the probability of the event $\{\tau < T\}$ is small: $\mathbb{P}(\tau < T) \leq \delta$. Note that

$$\{\tau < T\} = \{\tau_2 < T\} \cup \{\tau_3 < T\} \cup \{\tau_1 < T, \tau_2 = T, \tau_3 = T\}.$$

For any $i \in [K], j \in [3]$, we have that

$$\mathbb{P}(\|\varepsilon_{t,j,i}\| > \frac{L_0}{\sqrt{\alpha\rho}}) = \mathbb{P}(\|\varepsilon_{t,j,i}\|^2 > \frac{L_0^2}{\alpha\rho}) \leq \frac{\sigma^2\alpha\rho}{L_0^2},$$

where the last inequality is due to Chebyshev's inequality. Based on the union bound, we have that

$$\mathbb{P}(\{\tau_2 < T\}) \leq \sum_{t=0}^{T-1} \sum_{j=1}^{K} \sum_{i=1}^{3} \mathbb{P}(\|\varepsilon_{t,j,i}\| > \frac{L_0}{\sqrt{\alpha\rho}}) \leq \frac{3K\sigma^2\alpha\rho T}{L_0^2} \leq \frac{\delta}{8} \tag{15}$$

since $\rho \leq \frac{\delta L_0^2}{24K\sigma^2\alpha T}$. Similarly, we have that

$$\mathbb{P}(\|\varepsilon_{t,2,i}\|\|\varepsilon_{t,3,i}\| > \frac{L_1}{\sqrt{\alpha\rho}}) = \mathbb{P}(\|\varepsilon_{t,2,i}\|^2\|\varepsilon_{t,3,i}\|^2 > \frac{L_1^2}{\alpha\rho}) \leq \frac{\sigma^4\alpha\rho}{L_1^2}.$$

It follows that

$$\mathbb{P}(\{\tau_3 < T\}) \leq \sum_{t=0}^{T-1} \sum_{i=1}^{K} \sum_{j=1}^{K} \mathbb{P}(\|\varepsilon_{t,2,i}\|\|\varepsilon_{t,3,j}\| > \frac{L_1}{\sqrt{\alpha\rho}}) \leq \frac{K^2\sigma^4\alpha\rho T}{L_1^2} \leq \frac{\delta}{8}. \tag{16}$$

We then bound the probability of the event $\{\tau_1 < T, \tau_2 = T, \tau_3 = T\}$. Since $\tau = \tau_1 < T$, we have that for some $i \in [K], f_i(x_{\tau+1}) - f_i^* > F$.

According to equation 21 shown in Lemma 1, for any $i \in [K]$ and $t = \tau$ we have that

$$
\begin{aligned}
f_i(x_{\tau+1}) - f_i(x_\tau) &\leq \alpha \|\nabla F(x_\tau)w\| \|\varepsilon_{t,1}w_t\| + \ell(M+1)\alpha^2 \|\varepsilon_{t,1}w_t\|^2 + \frac{\alpha}{\beta} + \alpha\rho + \alpha\beta\rho^2 \\
&\quad + \alpha M \|\varepsilon_{t,2}\| + \alpha M \|\varepsilon_{t,3}\| + \alpha \|\varepsilon_{t,2}^\top \varepsilon_{t,3} w_t\| \\
&\quad + 4\alpha\beta K M^4 + 4\alpha\beta M^2 \|\varepsilon_{t,2}\|^2 + 4\alpha\beta M^2 \|\varepsilon_{t,3}\|^2 + 4\alpha\beta \|\varepsilon_{t,2}^\top \varepsilon_{t,3} w_t\|^2 \\
&\leq \alpha M \frac{L_0}{\sqrt{\alpha\rho}} + \ell(M+1)\alpha \frac{L_0^2}{\rho} + \frac{\alpha}{\beta} + \alpha\rho + \alpha\beta\rho^2 \\
&\quad + \alpha M \frac{\sqrt{K}L_0}{\sqrt{\alpha\rho}} + \alpha M \frac{\sqrt{K}L_0}{\sqrt{\alpha\rho}} + \alpha M \frac{\sqrt{K}L_1}{\sqrt{\alpha\rho}} \\
&\quad + 4\alpha\beta K M^4 + 4\alpha\beta M^2 \frac{KL_0^2}{\alpha\rho} + 4\alpha\beta M^2 \frac{KL_0^2}{\alpha\rho} + 4\alpha\beta \frac{KL_1^2}{\alpha\rho} \\
&\leq b_1 + b_2 + b_3 + c_1 + \alpha\rho(1 + \beta\rho) + 4\alpha\beta K M^4 \\
&\leq \frac{F}{2},
\end{aligned}
$$

where the first inequality is due to that $\tau_2 = \tau_3 = T$, and the second one is due to $\frac{\beta}{\rho} \leq \frac{b_3}{8KM^2L_0^2 + 4KL_1^2}$ and $\frac{\alpha}{\rho} \leq \min\left\{\frac{b_1^2}{(3M\sqrt{K}L_0 + M\sqrt{K}L_1)^2}, \frac{b_2}{\ell(M+1)L_0^2}\right\}$.

However, for some $i \in [K]$, $f_i(x_{\tau+1}) - f_i^* > F$. Thus for this task, we have that

$$
f_i(x_\tau) - f_i^* > \frac{F}{2}.
$$

According to Lemma 1, we have that

$$
\mathbb{E}[f_i(x_\tau) - f_i^*] \leq \frac{\delta}{8}F.
$$

Based on Markov inequality, it follows that

$$
\mathbb{P}\left(f_i(x_\tau) - f_i^* \leq \frac{F}{2}\right) \leq \frac{\mathbb{E}[f_i(x_\tau) - f_i^*]}{F/2} \leq \frac{\delta}{4}, \tag{17}
$$

which indicates that $\mathbb{P}(\tau_1 < T, \tau_2 = T, \tau_3 = T) \leq \frac{\delta}{4}$. It follows that $\mathbb{P}(\tau < T) \leq \frac{\delta}{2}$.

**Convergence of** $\frac{1}{T}\mathbb{E}\left[\sum_{t=0}^{T-1} \|\nabla F(x_t)w_t\|^2 \Big| \tau = T\right]$. Based on equation 24 in Lemma 1, we have that

$$
\begin{aligned}
&\frac{1}{T}\mathbb{E}\left[\sum_{t=0}^{T-1} \|\nabla F(x_t)w_t\|^2 \Big| \tau = T\right] \\
&\leq \frac{1}{T}\frac{1}{\mathbb{P}(\tau = T)}\mathbb{E}\left[\sum_{t=0}^{\tau-1} \|\nabla F(x_t)w_t\|^2\right] \\
&\leq \frac{4F(x_0)w - 4F^*w + \frac{4\alpha}{\beta}}{\alpha T} + \frac{4\sqrt{2}M(3\sigma + \sigma^2)}{\sqrt{T}} + 4\alpha\sigma^2\ell(M+1) + 4\rho \\
&\quad + 4\beta\rho^2 + 16\beta K M^4 + 32\beta K M^2\sigma^2 + 16\beta K\sigma^4 \\
&\leq \frac{\delta}{2}\epsilon^2,
\end{aligned}
$$

where the second inequality is due to $\delta < \frac{1}{2}$ and the last inequality is due to our selection of parameters. As a result, we have that

$$
\mathbb{P}\left(\frac{1}{T}\sum_{t=0}^{T-1} \|\nabla F(x_t)w_t\|^2 > \epsilon^2 \Big| \tau = T\right) \leq \frac{\mathbb{E}\left[\frac{1}{T}\sum_{t=0}^{T-1} \|\nabla F(x_t)w_t\|^2 \geq \epsilon^2 \Big| \tau = T\right]}{\epsilon^2} \leq \frac{\delta}{2}, \tag{18}
$$

where the first probability is due to Markov inequality. Thus we have that

$$\mathbb{P}\left(\frac{1}{T}\sum_{t=0}^{T-1}\|\nabla F(x_t)w_t\|^2 \le \epsilon^2\right)$$

$$\ge 1 - \mathbb{P}\left(\tau < T\right) - \mathbb{P}\left(\frac{1}{T}\sum_{t=0}^{T-1}\|\nabla F(x_t)w_t\|^2 > \epsilon^2 \Big| \tau = T\right)\mathbb{P}\left(\tau = T\right)$$

$$\ge 1 - \delta,$$

where the last inequality is due to equation 15, equation 16, equation 17, and equation 18. This completes the proof. $\square$

### D.4 Proof of Lemma 1

*Proof.* For all $i \in [K], t \le \tau$, we have $f_i(x_t) - f_i^* \le F$ which further implies that $\|\nabla f_i(x_t)\| \le M$. Moreover, we have that for any $t \le \tau$ and $i \in [K]$,

$$\|x_{t+1} - x_t\| \le \alpha\|\nabla G_1(x_t)w_t\| \le \alpha(\|\nabla F(x_t)w_t\| + \|\varepsilon_{t,1}w_t\|) \le \alpha\left(M + \frac{L_0}{\sqrt{\alpha\rho}}\right) \le \frac{1}{\ell(M+1)}.$$

Since $f_i(x)$ is $\left(\frac{1}{\ell(\|\nabla f_i(x))\|+1)}, \ell(\|\nabla f_i(x)\| + 1)\right)$-smooth, it follows that

$$f_i(x_{t+1}) - f_i(x_t) \le -\alpha\langle\nabla f_i(x_t), \nabla G_1(x_t)w_t\rangle + \frac{\ell(\|\nabla f_i(x_t)\|+1)}{2}\alpha^2\|\nabla G_1(x_t)w_t\|^2.$$

As a result, for any $w \in \mathcal{W}$, we have that

$$F(x_{t+1})w \le F(x_t)w - \alpha\langle\nabla F(x_t)w, \nabla G_1(x_t)w_t\rangle + \frac{\ell(M+1)}{2}\alpha^2\|\nabla G_1(x_t)w_t\|^2$$

$$\le F(x_t)w - \alpha\langle\nabla F(x_t)w, \nabla F(x_t)w_t\rangle + \alpha\langle\nabla F(x_t)w, \varepsilon_{t,1}w_t\rangle$$
$$+ \ell(M+1)\alpha^2\|\nabla F(x_t)w_t\|^2 + \ell(M+1)\alpha^2\|\varepsilon_{t,1}w_t\|^2 \qquad (19)$$

Based on the update process of $w$, we have that

$$\|w_{t+1} - w\|^2$$
$$= \|\Pi_{\mathcal{W}}\left(w_t - \beta[\nabla G_2(x_t)^\top\nabla G_3(x_t)w_t + \rho w_t]\right) - w\|^2$$
$$\le \|\left(w_t - \beta[\nabla G_2(x_t)^\top\nabla G_3(x_t)w_t + \rho w_t]\right) - w\|^2$$
$$= \|w_t - w\|^2 - 2\beta\langle w_t - w, (\nabla G_2(x_t)^\top\nabla G_3(x_t) + \rho)w_t\rangle$$
$$+ \beta^2\|(\nabla G_2(x_t)^\top\nabla G_3(x_t) + \rho)w_t\|^2,$$

where the inequality follows from the non-expansiveness of projection. It follows that

$$2\beta\langle w_t - w, \nabla F(x_t)^\top\nabla F(x_t)w_t\rangle$$
$$\le \left(\|w_t - w\|^2 - \|w_{t+1} - w\|^2\right) + 2\beta\rho + 2\beta^2\rho^2$$
$$+ 2\beta\left\langle w_t - w, \varepsilon_{t,2}^\top\nabla F(x_t)w_t + \nabla F(x_t)^\top\varepsilon_{t,3}w_t - \varepsilon_{t,2}^\top\varepsilon_{t,3}w_t\right\rangle$$
$$+ 2\beta^2\|\nabla F(x_t)^\top\nabla F(x_t)w_t - \varepsilon_{t,2}^\top\nabla F(x_t)w_t - \nabla F(x_t)^\top\varepsilon_{t,3}w_t + \varepsilon_{t,2}^\top\varepsilon_{t,3}w_t\|^2$$
$$\le \left(\|w_t - w\|^2 - \|w_{t+1} - w\|^2\right) + 2\beta\rho + 2\beta^2\rho^2$$
$$+ 2\beta\left\langle w_t - w, \varepsilon_{t,2}^\top\nabla F(x_t)w_t + \nabla F(x_t)^\top\varepsilon_{t,3}w_t - \varepsilon_{t,2}^\top\varepsilon_{t,3}w_t\right\rangle$$
$$+ 8\beta^2 KM^4 + 8\beta^2 M^2\|\varepsilon_{t,2}\|^2 + 8\beta^2 M^2\|\varepsilon_{t,3}\|^2 + 8\beta^2\|\varepsilon_{t,2}^\top\varepsilon_{t,3}w_t\|^2. \qquad (20)$$

Combine equation 19 and equation 20, and we can get that

$$F(x_{t+1})w - F(x_t)w \le -\alpha\|\nabla F(x_t)w_t\|^2 + \alpha\langle\nabla F(x_t)w, \varepsilon_{t,1}w_t\rangle$$
$$+ \ell(M+1)\alpha^2\|\nabla F(x_t)w_t\|^2 + \ell(M+1)\alpha^2\|\varepsilon_{t,1}w_t\|^2$$
$$+ \frac{\alpha}{2\beta}\left(\|w_t - w\|^2 - \|w_{t+1} - w\|^2\right) + \alpha\rho + \alpha\beta\rho^2$$
$$+ \alpha\left\langle w_t - w, \varepsilon_{t,2}^\top\nabla F(x_t)w_t + \nabla F(x_t)w_t^\top\varepsilon_{t,3} - \varepsilon_{t,2}^\top\varepsilon_{t,3}w_t\right\rangle$$
$$+ 4\alpha\beta KM^4 + 4\alpha\beta M^2\|\varepsilon_{t,2}\|^2 + 4\alpha\beta M^2\|\varepsilon_{t,3}\|^2 + 4\alpha\beta\|\varepsilon_{t,2}^\top\varepsilon_{t,3}w_t\|^2. \qquad (21)$$

Taking expectation and sum up equation 21 from $t = 0$ to $\tau - 1$, we have that

$$\mathbb{E}[F(x_\tau)w] - F(x_0)w \leq -\frac{\alpha}{2}\mathbb{E}\left[\sum_{t=0}^{\tau-1}\|\nabla F(x_t)w_t\|^2\right] + \alpha\mathbb{E}\left[\sum_{t=0}^{\tau-1}\langle\nabla F(x_t)w, \varepsilon_{t,1}w_t\rangle\right]$$

$$+ \alpha\mathbb{E}\left[\sum_{t=0}^{\tau-1}\langle w_t - w, \varepsilon_{t,2}^\top\nabla F(x_t)w_t + \nabla F(x_t)w_t^\top\varepsilon_{t,3} - \varepsilon_{t,2}^\top\varepsilon_{t,3}w_t\rangle\right]$$

$$+ \ell(M+1)\alpha^2\mathbb{E}\left[\sum_{t=0}^{\tau-1}\|\varepsilon_{t,1}w_t\|^2\right] + \frac{\alpha}{2\beta}\|w_0 - w\|^2 + \alpha\rho T + \alpha\beta\rho^2 T$$

$$+ 4\alpha\beta KM^4 T + 4\alpha\beta M^2\mathbb{E}\left[\sum_{t=0}^{\tau-1}\|\varepsilon_{t,2}\|^2\right]$$

$$+ 4\alpha\beta M^2\mathbb{E}\left[\sum_{t=0}^{\tau-1}\|\varepsilon_{t,3}\|^2\right] + 4\alpha\beta\mathbb{E}\left[\sum_{t=0}^{\tau-1}\|\varepsilon_{t,2}^\top\varepsilon_{t,3}w_t\|^2\right]$$

$$\leq -\frac{\alpha}{2}\mathbb{E}\left[\sum_{t=0}^{\tau-1}\|\nabla F(x_t)w_t\|^2\right] + \alpha\mathbb{E}\left[\sum_{t=0}^{\tau-1}\langle\nabla F(x_t)w, \varepsilon_{t,1}w_t\rangle\right]$$

$$+ \alpha\mathbb{E}\left[\sum_{t=0}^{\tau-1}\langle w_t - w, \varepsilon_{t,2}^\top\nabla F(x_t)w_t + \nabla F(x_t)w_t^\top\varepsilon_{t,3} - \varepsilon_{t,2}^\top\varepsilon_{t,3}w_t\rangle\right]$$

$$+ \ell(M+1)\alpha^2 T\sigma^2 + \frac{\alpha}{\beta} + \alpha\rho T + \alpha\beta\rho^2 T$$

$$+ 4\alpha\beta KM^4 T + 4\alpha\beta KM^2 T\sigma^2$$

$$+ 4\alpha\beta KM^2 T\sigma^2 + 4\alpha\beta TK\sigma^4, \tag{22}$$

where the last inequality is due to that $\tau \leq T$ and for any $i \in [K], j \in [3]$, $\mathbb{E}\left[\sum_{t=0}^{\tau-1}\|\varepsilon_{t,j,i}\|^2\right] \leq \mathbb{E}\left[\sum_{t=0}^{T-1}\|\varepsilon_{t,j,i}\|^2\right] \leq TK\sigma^2$. By the optional stopping theorem, we have that

$$\mathbb{E}\left[\sum_{t=0}^{\tau}\langle\nabla F(x_t)w, \varepsilon_{t,1}w_t\rangle\right] = 0,$$

which further implies that

$$\mathbb{E}\left[\sum_{t=0}^{\tau-1}\langle\nabla F(x_t)w, \varepsilon_{t,1}w_t\rangle\right] = -\mathbb{E}[\langle\nabla F(x_\tau)w, \varepsilon_{\tau,1}w_\tau\rangle]$$

$$\leq \mathbb{E}[M\|\varepsilon_{\tau,1}w_\tau\|] \leq M\sqrt{\mathbb{E}[\|\varepsilon_{\tau,1}w_\tau\|^2]}$$

$$\leq M\sqrt{\mathbb{E}\left[\sum_{t=0}^{T}\|\varepsilon_{t,1}w_t\|^2\right]} \leq M\sigma\sqrt{T+1}$$

$$\leq \sqrt{2}M\sigma\sqrt{T}. \tag{23}$$

Similarly, we have $\mathbb{E}\left[\sum_{t=0}^{\tau-1}\langle\nabla F(x_t)w, \varepsilon_{t,2}w_t\rangle\right] \leq \sqrt{2}M\sigma\sqrt{T}$, $\mathbb{E}\left[\sum_{t=0}^{\tau-1}\langle\nabla F(x_t)w, \varepsilon_{t,3}w_t\rangle\right] \leq \sqrt{2}M\sigma\sqrt{T}$ and $\mathbb{E}\left[\sum_{t=0}^{\tau-1}\langle\nabla F(x_t)w, \varepsilon_{t,2}^\top\varepsilon_{t,3}w_t\rangle\right] \leq \sqrt{2K}M\sigma^2\sqrt{T}$.

Based on equation 22 and equation 23, we have that

$$\mathbb{E}[F(x_\tau)w] - F^*w \le F(x_0)w - F^*w - \mathbb{E}\left[\sum_{t=0}^{\tau-1} \frac{\alpha}{2}\|\nabla F(x_t)w_t\|^2\right] + \alpha\sqrt{2T}M(3\sigma + \sigma^2)$$
$$+ \ell(M+1)\alpha^2 T\sigma^2 + \frac{\alpha}{\beta} + \alpha\rho T + \alpha\beta\rho^2 T$$
$$+ 4\alpha\beta KM^4 T + 4\alpha\beta KM^2 T\sigma^2$$
$$+ 4\alpha\beta KM^2 T\sigma^2 + 4\alpha\beta TK\sigma^4$$
$$\le \frac{\delta F}{8} - \mathbb{E}\left[\sum_{t=0}^{\tau-1} \frac{\alpha}{2}\|\nabla F(x_t)w_t\|^2\right], \tag{24}$$

which completes the proof. $\qquad\square$

## E    DETAILED PROOFS FOR ITERATION-WISE CA DISTANCE

We first provide some useful lemmas, which will be used in our main theorems.

**Lemma 3** (Continuity of $w_{t,\rho}^*$). *Suppose Assumptions 1 and 2 are satisfied. If for any $i \in [K], \|\nabla f_i(x_t)\| \le M$ and $\|x_t - x_{t+1}\| \le \frac{1}{\ell(M+1)}$, we have,*

$$\|w_\rho^*(x_t) - w_\rho^*(x_{t+1})\| \le L_w\|x_t - x_{t+1}\|,$$

*where $L_w = 2\rho^{-1}KM\ell(M+1)$.*

*Proof.* We first define that $w_{Q,\rho}(x_t) \in \mathcal{W}$ is the $Q$-th iterate of a function $J(w) = \frac{1}{2}\|\nabla F(x_t)w\|^2 + \frac{\rho}{2}\|w\|^2$ using projected gradient descent (PGD) with a constant step size $\beta$. The update rule is $w_{Q+1,\rho}(x_t) = \Pi_{\mathcal{W}}\left(((1-\beta\rho)I - \beta\nabla F(x_t)^\top\nabla F(x_t))w_{Q,\rho}(x_t)\right)$. By the non-expansiveness of projection, we have

$$\|w_{Q+1,\rho}(x_t) - w_{Q+1,\rho}(x_{t+1})\|$$
$$\le\|\left((1-\beta\rho)I - \beta\nabla F(x_t)^\top\nabla F(x_t)\right)w_{Q,\rho}(x_t)$$
$$- \left((1-\beta\rho)I - \beta\nabla F^\top(x_{t+1})\nabla F(x_{t+1})\right)w_{Q,\rho}(x_{t+1})\|$$
$$\le\|(1-\beta\rho)I - \beta\nabla F(x_t)^\top\nabla F(x_t)\|\|w_{Q,\rho}(x_t) - w_{Q,\rho}(x_{t+1})\|$$
$$+ \beta\|\left(\nabla F(x_t)^\top\nabla F(x_t) - \nabla F^\top(x_{t+1})\nabla F(x_{t+1})\right)w_{Q,\rho}(x_{t+1})\|$$
$$\le(1-\beta\rho)\|w_{Q,\rho}(x_t) - w_{Q,\rho}(x_{t+1})\|$$
$$+ \beta\|\left(\nabla F(x_t)^\top\nabla F(x_t) - \nabla F^\top(x_{t+1})\nabla F(x_{t+1})\right)w_{Q,\rho}(x_{t+1})\|.$$

Since we set $w_{0,\rho}(x_t) = w_{0,\rho}(x_{t+1})$ and $\|w_{Q,\rho}(x_{t+1})\| \le 1$, telescoping the above inequality over $Q$ gives,

$$\|w_{Q,\rho}(x_t) - w_{Q,\rho}(x_{t+1})\| \le \rho^{-1}\|\nabla F(x_t)^\top\nabla F(x_t) - \nabla F^\top(x_{t+1})\nabla F(x_{t+1})\|. \tag{25}$$

Then according to the Cauchy-Schwartz inequality, it follows that

$$\|w_\rho^*(x_t) - w_\rho^*(x_{t+1})\| \le \lim_{Q\to\infty}\left(\|w_\rho^*(x_t) - w_{Q,\rho}(x_t)\| + \|w_\rho^*(x_{t+1}) - w_{Q,\rho}(x_{t+1})\|\right.$$
$$\left. + \|w_{Q,\rho}(x_t) - w_{Q,\rho}(x_{t+1})\|\right)$$
$$\overset{(i)}{\le} \lim_{Q\to\infty}\left(\|w_\rho^*(x_t) - w_{Q,\rho}(x_t)\| + \|w_\rho^*(x_{t+1}) - w_{Q,\rho}(x_{t+1})\|\right)$$
$$+ \rho^{-1}\|\nabla F(x_t)^\top\nabla F(x_t) - \nabla F^\top(x_{t+1})\nabla F(x_{t+1})\|$$
$$\overset{(ii)}{\le} \lim_{Q\to\infty} 2\sqrt{\frac{4}{\rho\beta Q}} + \rho^{-1}\|\nabla F(x_t)^\top\nabla F(x_t) - \nabla F^\top(x_{t+1})\nabla F(x_{t+1})\|, \tag{26}$$

where $(i)$ follows from eq. (25) and $(ii)$ follows from the convergence of PGD (Theorem 1.1, (Beck & Teboulle, 2009)) on $\rho$-strongly convex objectives that

$$\|w_\rho^*(x_t) - w_{Q,\rho}(x_t)\|^2 \leq \frac{2}{\rho}\big(J(w_\rho^*(x_t)) - J(w_{Q,\rho}(x_t))\big) \leq \frac{2}{\rho}\frac{\|w_{0,\rho}(x_t) - w_\rho^*(x_t)\|^2}{2\beta Q} \leq \frac{4}{\rho\beta Q}.$$

Then eq. (26) can be bounded by

$$\begin{aligned}\|w_\rho^*(x_t) - w_\rho^*(x_{t+1})\| &\leq \rho^{-1}\|\nabla F(x_t)^\top \nabla F(x_t) - \nabla F^\top(x_{t+1})\nabla F(x_{t+1})\| \\ &\leq \rho^{-1}\|\nabla F(x_t) + \nabla F(x_{t+1})\|\|\nabla F(x_t) - \nabla F(x_{t+1})\| \\ &\leq 2\rho^{-1}KM\ell(M+1)\|x_t - x_{t+1}\|,\end{aligned}$$

where the last inequality follows from $\|\nabla f_i(x_t)\| \leq M$ and $f_i(x)$ is $\Big(\frac{1}{\ell(\|\nabla f_i(x)\|+1)}, \ell(\|\nabla f_i(x)\| + 1)\Big)$-smooth by setting $a = 1$. The proof is complete. $\qquad\square$

**Lemma 4.** *Given $w_t^* = \arg\min_{w\in\mathcal{W}}\frac{1}{2}\|\nabla F(x_t)w\|^2$ and $w_{t,\rho}^* = \arg\min_{w\in\mathcal{W}}\frac{1}{2}\|\nabla F(x_t)w\|^2 + \frac{\rho}{2}\|w\|^2$, we have*

$$\|\nabla F(x_t)w_t^* - \nabla F(x_t)w_{t,\rho}^*\| \leq \sqrt{\rho}.$$

*Proof.* Recall that $w_{t,\rho}^* = \arg\min_{w\in\mathcal{W}}\frac{1}{2}\|\nabla F(x_t)w\|^2 + \frac{\rho}{2}\|w\|^2$, then we have

$$\frac{1}{2}\|\nabla F(x_t)w_t^*\|^2 + \frac{\rho}{2}\|w_t^*\|^2 - \frac{1}{2}\|\nabla F(x_t)w_{t,\rho}^*\|^2 - \frac{\rho}{2}\|w_{t,\rho}^*\|^2 \geq 0.$$

By rearranging the above inequality, we have

$$\|\nabla F(x_t)w_{t,\rho}^*\|^2 - \|\nabla F(x_t)w_t^*\|^2 \leq \rho(\|w_{t,\rho}^*\|^2 - \|w_t^*\|^2) \leq \rho.$$

Then recall that $w_t^* = \arg\min_{w\in\mathcal{W}}\frac{1}{2}\|\nabla F(x_t)w\|^2$, we have

$$\begin{aligned}\|\nabla F(x_t)w_t^* - \nabla F(x_t)w_{t,\rho}^*\|^2 &= \|\nabla F(x_t)w_t^*\|^2 + \|\nabla F(x_t)w_{t,\rho}^*\|^2 - 2\langle\nabla F(x_t)w_t^*, \nabla F(x_t)w_{t,\rho}^*\rangle \\ &\leq \|\nabla F(x_t)w_{t,\rho}^*\|^2 - \|\nabla F(x_t)w_t^*\|^2 \\ &\leq \rho,\end{aligned}$$

where the first inequlity follows from the optimality that

$$-2\langle w_{t,\rho}^*, \nabla F(x_t)^\top \nabla F(x_t)w_t^*\rangle \leq -2\|\nabla F(x_t)w_t^*\|^2.$$

The proof is complete. $\qquad\square$

**Lemma 5.** *Suppose Assumptions 1 and 2 are satisfied. If for any $i \in [K]$, $\|\nabla f_i(x_t)\| \leq M$ and $\|x_t - x_{t+1}\| \leq \frac{1}{\ell(M+1)}$, we have*

$$\|R(x_t)\| \leq \frac{\alpha^2 K\ell(M+1)M^2}{2}.$$

*Proof.* According to the Talyor Theorem, we have the following result for any objective function $f_i(x_t), i \in [K]$.

$$f_i(x_{t+1}) = f_i(x_t) + \nabla f_i^\top(x_t)(x_{t+1} - x_t) + R_i(x_t),$$

where $R_i(x_t)$ is the remainder term. Then according to the descent lemma of each objective function $f_i(x)$, we have

$$\begin{aligned}f_i(x_{t+1}) &\leq f_i(x_t) + \nabla f_i^\top(x_t)(x_{t+1} - x_t) + \alpha^2\frac{\ell(\|\nabla f_i(x_t)\| + 1)}{2}\|\nabla F(x_t)w_t\|^2 \\ &\leq f_i(x_t) + \nabla f_i^\top(x_t)(x_{t+1} - x_t) + \alpha^2\frac{\ell(M+1)}{2}\|\nabla F(x_t)w_t\|^2.\end{aligned}$$

Then we can obtain

$$R_i(x_t) \leq \alpha^2\frac{\ell(M+1)}{2}\|\nabla F(x_t)w_t\|^2.$$

Thus, according to the Cauchy-Schwartz inequality, we have

$$\|R(x_t)\| \leq \alpha^2\frac{\ell(M+1)}{2}\|\nabla F(x_t)w_t\|^2 \leq \frac{\alpha^2 K\ell(M+1)M^2}{2}.$$

The proof is complete. $\qquad\square$

### E.1 PROOF OF THEOREM 3

**Theorem 9.** *Suppose Assumptions 1 and 2 are satisfied. We choose* $\beta' \leq \frac{1}{M^2}, N = \mathcal{O}(\epsilon^{-2}), C_1 \geq \sqrt{K}M^2 + \rho, \beta \leq \min\left(\frac{\epsilon^2\rho}{C_1^2}, \epsilon^2, \frac{1}{4KM^2}\right), \alpha \leq \min\left(\beta, \frac{1}{\ell(M+1)}, \frac{\beta\rho\epsilon}{2L_w\sqrt{M}}, \frac{\rho\epsilon^2}{2L_wMC_1}\right), T \geq \max\left(\frac{10\Delta}{\alpha\epsilon^2}, \frac{10}{\epsilon^2\beta}\right) = \Theta(\epsilon^{-11}),$ *and* $\rho \leq \min\left(\frac{\epsilon^2}{20}, \frac{1}{2T\alpha}, \sqrt{\frac{\epsilon^2}{10\beta}}, \sqrt{\frac{1}{T\alpha\beta}}\right) = \mathcal{O}(\epsilon^2).$ *The CA distance in every iteration takes the order of* $\mathcal{O}(\epsilon).$

*Proof.* Since our parameters satisfy all requirements in Theorem 1, we have that $\|\nabla f_i(x_t)\| \leq M$. According to the definition of CA distance, we have

$$
\begin{aligned}
&\|\nabla F(x_t)w_t - \nabla F(x_t)w_t^*\| \\
&= \|\nabla F(x_t)w_t - \nabla F(x_t)w_{t,\rho}^* + \nabla F(x_t)w_{t,\rho}^* - \nabla F(x_t)w_t^*\| \\
&\overset{(i)}{\leq} \|\nabla F(x_t)w_t - \nabla F(x_t)w_{t,\rho}^*\| + \|\nabla F(x_t)w_{t,\rho}^* - \nabla F(x_t)w_t^*\| \\
&\overset{(ii)}{\leq} \sqrt{K}M\|w_t - w_{t,\rho}^*\| + \sqrt{\rho},
\end{aligned}
\tag{27}
$$

where $(i)$ follows from Cauchy-Schwartz inequality and $(ii)$ follows from $\|\nabla f_i(x_t)\| \leq M$ for any $i$ and Lemma 4. Then for the first term in the above inequality on the right-hand side (RHS), we have

$$
\|w_{t+1} - w_{t+1,\rho}^*\|^2 = \|w_{t+1} - w_{t,\rho}^*\|^2 + \|w_{t+1,\rho}^* - w_{t,\rho}^*\|^2 - 2\langle w_{t+1} - w_{t,\rho}^*, w_{t+1,\rho}^* - w_{t,\rho}^*\rangle.
\tag{28}
$$

For the first term on the RHS in the above inequality, we have

$$
\begin{aligned}
\|w_{t+1} - w_{t,\rho}^*\|^2 &\overset{(i)}{\leq} \|w_t - \beta[\nabla F(x_t)^\top \nabla F(x_t)w_t + \rho w_t] - w_{t,\rho}^*\|^2 \\
&= \|w_t - w_{t,\rho}^*\|^2 - 2\beta\langle \nabla F(x_t)^\top \nabla F(x_t)w_t + \rho w_t, w_t - w_{t,\rho}^*\rangle \\
&\quad + \beta^2\|\nabla F(x_t)^\top \nabla F(x_t)w_t + \rho w_t\|^2 \\
&\overset{(ii)}{\leq} (1 - 2\beta\rho)\|w_t - w_{t,\rho}^*\|^2 + \beta^2(\rho + \sqrt{K}M^2)^2,
\end{aligned}
\tag{29}
$$

where $(i)$ follows from the non-expansiveness of projection and $(ii)$ follows from properties of strong convexity and Cauchy-Schwartz inequality. Then for the second term on the RHS in eq. (28), we have

$$
\|w_{t+1,\rho}^* - w_{t,\rho}^*\|^2 \leq L_w^2\|x_t - x_{t+1}\|^2 = L_w^2\alpha^2\|\nabla F(x_t)w_t\|^2 \leq \alpha^2 L_w^2 M^2.
\tag{30}
$$

Then for the last term on the RHS in eq. (28), we have

$$
\begin{aligned}
&-2\langle w_{t+1} - w_{t,\rho}^*, w_{t+1,\rho}^* - w_{t,\rho}^*\rangle \\
&\leq 2\|w_{t+1} - w_{t,\rho}^*\|\|w_{t+1,\rho}^* - w_{t,\rho}^*\| \\
&\leq 2(\|w_{t+1} - w_t\| + \|w_t - w_{t,\rho}^*\|)\|w_{t+1,\rho}^* - w_{t,\rho}^*\| \\
&\overset{(i)}{\leq} 2\alpha\beta L_w\|\nabla F(x_t)^\top \nabla F(x_t)w_t + \rho w_t\|\|\nabla F(x_t)w_t\| + \beta\rho\|w_t - w_{t,\rho}^*\|^2 + \frac{4}{\beta\rho}\|w_{t+1,\rho}^* - w_{t,\rho}^*\|^2 \\
&\leq 2\alpha\beta L_w M(\sqrt{K}M^2 + \rho) + \beta\rho\|w_t - w_{t,\rho}^*\|^2 + \frac{4\alpha^2 L_w^2 M^2}{\beta\rho},
\end{aligned}
\tag{31}
$$

where $(i)$ follows from the update rule in Algorithm 1, Lemma 3, and Young's inequality. Then substituting eq. (29), eq. (30) and eq. (31) into eq. (28), we have

$$
\begin{aligned}
\|w_{t+1} - w_{t+1,\rho}^*\|^2 &\leq (1 - \beta\rho)\|w_t - w_{t,\rho}^*\|^2 + \beta^2(\rho + \sqrt{K}M^2)^2 + \alpha^2 L_w^2 M^2 \\
&\quad + 2\alpha\beta L_w M(\sqrt{K}M^2 + \rho) + \frac{4\alpha^2 L_w^2 M^2}{\beta\rho} \\
&\leq (1 - \beta\rho)\|w_t - w_{t,\rho}^*\|^2 + \beta^2 C_1^2 + \alpha^2 L_w^2 M^2 + 2\alpha\beta L_w M C_1 + \frac{4\alpha^2 L_w^2 M^2}{\beta\rho},
\end{aligned}
$$

where the last inequality follows from Lemma 5 and $C_1 \geq \sqrt{K}M^2 + \rho$. Then we do telescoping over $t = 0, 1, ..., T - 1$

$$\|w_T - w^*_{T,\rho}\|^2 \leq (1 - \beta\rho)^T \|w_0 - w^*_{0,\rho}\|^2 + \frac{\beta}{\rho}C_1^2 + \frac{\alpha^2}{\beta\rho}L_w^2 M^2 + \frac{2\alpha L_w M}{\rho}C_1 + \frac{4\alpha^2 L_w^2 M^2}{\beta^2\rho^2}.$$

Then recalling that $L_w = \mathcal{O}(\frac{1}{\rho})$ and substituting the above inequality into eq. (27), we have

$$\|\nabla F(x_t)w_t - \nabla F(x_t)w^*_t\|$$
$$\leq \sqrt{K}M\Big[(1 - \beta\rho)^t \|w_0 - w^*_{0,\rho}\|^2 + \frac{\beta}{\rho}C_1^2 + \frac{\alpha^2}{\beta\rho}L_w^2 M +$$
$$\frac{2\alpha L_w M}{\rho}C_1 + \frac{4\alpha^2 L_w^2 M^2}{\beta^2\rho^2}\Big]^{\frac{1}{2}} + \sqrt{\rho}$$
$$= \mathcal{O}\Big((1 - \beta\rho)^{\frac{t}{2}} \|w_0 - w^*_{0,\rho}\| + \sqrt{\frac{\beta}{\rho}} + \frac{\alpha}{\beta\rho^2} + \sqrt{\rho}\Big).$$

Since we run projected gradient descent for the strongly convex function $J(w_n) = \frac{1}{2}\|\nabla F(x_0)w_n\|^2 + \frac{\rho}{2}\|w_n\|^2$ in the N-loop in Algorithm 1, according to Theorem 10.5 (Garrigos & Gower, 2023), we have by choosing $\beta' \in (0, \frac{1}{M^2}]$

$$\|w_0 - w^*_{0,\rho}\|^2 = \|w_N - w^*_{0,\rho}\|^2 \leq 2\Big(1 - \frac{\rho}{M^2}\Big)^N.$$

Thus, $\|w_0 - w^*_{0,\rho}\| = \mathcal{O}(\epsilon)$ as $N = \mathcal{O}(\rho^{-1})$. CA distance takes the order of $\epsilon$ in every iteration by choosing $\rho = \mathcal{O}(\epsilon^2), \beta = \mathcal{O}(\epsilon^4), \alpha = \mathcal{O}(\epsilon^9)$, and $N = \mathcal{O}(\epsilon^{-2})$. The proof is complete. $\qquad\square$

### E.2 FORMAL VERSION AND ITS PROOF OF THEOREM 4

Let $c_1 > 0, c'_2 > 0, c'_3, c'_4 \geq 0$, and $F > 0$ be some constants such that

$$\Delta + c_1 + c'_2 + c'_3 + c'_4 \leq F$$

and $C'_1 \geq \sqrt{K}M^2 + \rho + \frac{\alpha K\ell(M+1)M^2}{2}$. We then have the following convergence rate for Algorithm 4.

**Theorem 10.** *Suppose Assumptions 1 and 2 are satisfied, and we choose constant step sizes that* $\beta \leq \frac{\epsilon^2}{C_1'^2}, \alpha \leq \min\Big(c_1\beta, \sqrt{\frac{2c'_3}{K\ell(M+1)M^2T}}, \frac{2c'_4}{\beta C_1'^2 T}, \frac{\epsilon^2}{K\ell(M+1)M^2}\Big), \rho \leq \min\Big(\frac{\epsilon^2}{2}, \frac{c'_2}{\alpha T}\Big),$ *and* $T \geq \max\Big(\frac{10\Delta}{\alpha\epsilon^2}, \frac{10}{\epsilon^2\beta}\Big)$. *We have*

$$\frac{1}{T}\sum_{t=0}^{T-1}\|\nabla F(x_t)w_t\|^2 = \mathcal{O}(\epsilon^2).$$

*Proof.* Following similar steps in Appendix D.1, we also prove that for any $i \in K$ and $t \leq T$, we have that $f_i(x_t) - f_i^* \leq F$ by induction.

**Base case:** since all constants $c_1, c'_2, c'_3, c'_4$ are non-negative, we have that $f_i(x_0) - f_i^* \leq \Delta \leq F$ holds for any $i \in [K]$.

**Induction step:** assume that for any $i \in [K]$ and $t \leq k < T$, $f_i(x_t) - f_i^* \leq F$ holds. We then prove $f_i(x_{k+1}) - f_i^* \leq F$ holds for any $i \in [K]$. Following similar steps in Appendix D.1, we have

$$F(x_{t+1})w \leq F(x_t)w - \alpha\langle\nabla F(x_t)w, \nabla F(x_t)w_t\rangle + \frac{\alpha^2\ell(M+1)}{2}\|\nabla F(x_t)w_t\|^2. \qquad (32)$$

Based on the update rule of $w$ and non-expansiveness of projection, we have

$$
\begin{aligned}
\|w_{t+1} - w\|^2 \leq & \left\|w_t - \beta\left(\frac{F(x_t) - F(x_{t+1})}{\alpha} + \rho w_t\right) - w\right\|^2 \\
= & \left\|w_t - \beta\left(\nabla F(x_t)^\top \nabla F(x_t)w_t + \rho w_t + \frac{R(x_t)}{\alpha}\right) - w\right\|^2 \\
\overset{(i)}{\leq} & \|w_t - w\|^2 - 2\beta\langle w_t - w, (\nabla F(x_t)^\top \nabla F(x_t) + \rho I)w_t\rangle + 2\frac{\beta}{\alpha}\|R(x_t)\| + \beta^2(C_1')^2 \\
\overset{(ii)}{\leq} & \|w_t - w\|^2 - 2\beta\langle w_t - w, (\nabla F(x_t)^\top \nabla F(x_t) + \rho I)w_t\rangle \\
& + \alpha\beta K\ell(M+1)M^2 + \beta^2(C_1')^2,
\end{aligned}
$$

where $(i)$ follows from Cauchy-Schwartz inequality and $C_1' \geq \sqrt{K}M^2 + \rho + \frac{\alpha K\ell(M+1)M^2}{2}$, and $(ii)$ follows from Lemma 5. Then we have

$$
\begin{aligned}
\langle w_t - w, \nabla F(x_t)^\top \nabla F(x_t)w_t\rangle \leq & \frac{1}{2\beta}(\|w_t - w\|^2 - \|w_{t+1} - w\|^2) + \rho \\
& + \frac{\alpha K\ell(M+1)M^2}{2} + \frac{\beta(C_1')^2}{2}.
\end{aligned}
$$

Then substituting the above inequality into eq. (32), we can obtain

$$
\begin{aligned}
F(x_{t+1})w - F(x_t) \leq & - \alpha\|\nabla F(x_t)w_t\|^2 + \frac{\alpha^2\ell(M+1)}{2}\|\nabla F(x_t)w_t\|^2 \\
& + \frac{\alpha}{2\beta}(\|w_t - w\|^2 - \|w_{t+1} - w\|^2) + \alpha\rho + \frac{\alpha^2 K\ell(M+1)M^2}{2} + \frac{\alpha\beta(C_1')^2}{2}.
\end{aligned}
$$

Then taking sums of the above inequality from $t = 0$ to $k$, for any $w \in \mathcal{W}$, we have

$$
\begin{aligned}
F(x_{k+1})w - F(x_0)w \leq & - \sum_{t=0}^{k}\alpha\|\nabla F(x_t)w_t\|^2 + \sum_{t=0}^{k}\frac{\alpha^2\ell(M+1)}{2}\|\nabla F(x_t)w_t\|^2 \\
& + \frac{\alpha}{2\beta}\|w_0 - w\|^2 + \alpha\rho T + \frac{\alpha^2 K\ell(M+1)M^2 T}{2} + \frac{\alpha\beta(C_1')^2 T}{2} \\
\leq & \frac{\alpha}{\beta} + \alpha\rho T + \frac{\alpha^2 K\ell(M+1)M^2 T}{2} + \frac{\alpha\beta(C_1')^2 T}{2}, \quad (33)
\end{aligned}
$$

where the last inequality follows from $\alpha \leq \frac{1}{\ell(M+1)}$. Thus, for any $i \in [K]$, it can be shown that

$$
f_i(x_{k+1}) - f_i^* \leq f_i(x_0) - f_i^* + \frac{\alpha}{\beta} + \alpha\rho T + \frac{\alpha^2 K\ell(M+1)M^2 T}{2} + \frac{\alpha\beta(C_1')^2 T}{2} \leq F,
$$

since we have that $\frac{\alpha}{\beta} \leq c_1$, $\alpha\rho T \leq c_2'$, $\frac{\alpha^2 K\ell(M+1)M^2 T}{2} \leq c_3'$, $\frac{\alpha\beta(C_1')^2 T}{2} \leq c_4'$. Now we finish the induction step and can show that $f_i(x_k) - f_i^* \leq F$ and eq. (33) hold for all $k < T$ and $i \in [K]$. Specifically, for $\alpha \leq \frac{1}{\ell(M+1)}$, we have

$$
\frac{1}{T}\sum_{t=0}^{T-1}\|\nabla F(x_t)w_t\|^2 \leq \frac{2F(x_0)w - 2F^*w}{\alpha T} + \frac{2}{\beta T} + 2\rho + \alpha K\ell(M+1)M^2 + \beta(C_1')^2.
$$

Then following the choice of step sizes, we can obtain

$$
\frac{1}{T}\sum_{t=0}^{T-1}\|\nabla F(x_t)w_t\|^2 = \mathcal{O}(\epsilon^2).
$$

The proof is complete. $\qquad\square$

### E.3 FORMAL VERSION AND ITS PROOF OF THEOREM 5

**Theorem 11.** *Suppose Assumptions 1 and 2 are satisfied. We choose* $\beta' \leq \frac{1}{M^2}, N = \Omega(\epsilon^{-2}), C_1' \geq \sqrt{K}M^2 + \rho + \frac{\alpha K \ell(M+1)M^2}{2}, \beta \leq \min\left(\frac{\epsilon^2 \rho}{(C_1')^2}, \epsilon^2\right), \alpha \leq \min\left(c_1\beta, \frac{2c_3'}{\beta c_1' T}, \frac{1}{\ell(M+1)}, \frac{\beta \rho \epsilon}{2L_w M}, \frac{\rho \epsilon^2}{2L_w MC_1'}\right), T \geq \max\left(\frac{10\Delta}{\alpha \epsilon^2}, \frac{10}{\epsilon^2 \beta}\right) = \Theta(\epsilon^{-11}),$ *and* $\rho \leq \min\left(\frac{\epsilon^2}{20}, \frac{c_2'}{2T\alpha}\right) = \mathcal{O}(\epsilon^2)$. *The CA distance in every iteration takes the order of* $\mathcal{O}(\epsilon)$.

*Proof.* According to the definition of CA distance, we have

$$\|\nabla F(x_t)w_t - \nabla F(x_t)w_t^*\|$$
$$=\|\nabla F(x_t)w_t - \nabla F(x_t)w_{t,\rho}^* + \nabla F(x_t)w_{t,\rho}^* - \nabla F(x_t)w_t^*\|$$
$$\overset{(i)}{\leq}\|\nabla F(x_t)w_t - \nabla F(x_t)w_{t,\rho}^*\| + \|\nabla F(x_t)w_{t,\rho}^* - \nabla F(x_t)w_t^*\|$$
$$\overset{(ii)}{\leq}\sqrt{K}M\|w_t - w_{t,\rho}^*\| + \sqrt{\rho}, \tag{34}$$

where $(i)$ follows from Cauchy-Schwartz inequality and $(ii)$ follows from $\|\nabla f_i(x_t)\| \leq M$ for any $i$ and Lemma 3. Then for the first term in the above inequality on the right-hand side (RHS), we have

$$\|w_{t+1} - w_{t+1,\rho}^*\|^2 = \|w_{t+1} - w_{t,\rho}^*\|^2 + \|w_{t+1,\rho}^* - w_{t,\rho}^*\|^2 - 2\langle w_{t+1} - w_{t,\rho}^*, w_{t+1,\rho}^* - w_{t,\rho}^*\rangle. \tag{35}$$

For the first term on the RHS in the above inequality, we have

$$\|w_{t+1} - w_{t,\rho}^*\|^2 \overset{(i)}{\leq} \left\|w_t - \beta\left(\frac{F(x_t) - F(x_{t+1})}{\alpha} + \rho w_t\right) - w_{t,\rho}^*\right\|^2$$
$$=\left\|w_t - \beta\left(\nabla F(x_t)^\top \nabla F(x_t)w_t + \rho w_t + \frac{R(x_t)}{\alpha}\right) - w_{t,\rho}^*\right\|^2$$
$$=\|w_t - w_{t,\rho}^*\|^2 - 2\beta\langle \nabla F(x_t)^\top \nabla F(x_t)w_t + \rho w_t, w_t - w_{t,\rho}^*\rangle$$
$$\quad - 2\frac{\beta}{\alpha}\langle R(x_t), w_t - w_{t,\rho}^*\rangle + \beta^2\left\|\nabla F(x_t)^\top \nabla F(x_t)w_t + \frac{R(x_t)}{\alpha} + \rho w_t\right\|^2$$
$$\overset{(ii)}{\leq}(1 - 2\beta\rho)\|w_t - w_{t,\rho}^*\|^2 + 2\frac{\beta}{\alpha}\|R(x_t)\| + \beta^2\left(\rho + \sqrt{K}M^2 + \frac{\|R(x_t)\|}{\alpha}\right)^2, \tag{36}$$

where $(i)$ follows from the non-expansiveness of projection and $(ii)$ follows from properties of strong convexity and Cauchy-Schwartz inequality. Then for the second term on the RHS in eq. (35), we have

$$\|w_{t+1,\rho}^* - w_{t,\rho}^*\|^2 \leq L_w^2\|x_t - x_{t+1}\|^2 = L_w^2\alpha^2\|\nabla F(x_t)w_t\|^2 \leq \alpha^2 L_w^2 M^2. \tag{37}$$

Then for the last term on the RHS in eq. (35), we have

$$-2\langle w_{t+1} - w_{t,\rho}^*, w_{t+1,\rho}^* - w_{t,\rho}^*\rangle$$
$$\leq 2\|w_{t+1} - w_{t,\rho}^*\|\|w_{t+1,\rho}^* - w_{t,\rho}^*\|$$
$$\leq 2(\|w_{t+1} - w_t\| + \|w_t - w_{t,\rho}^*\|)\|w_{t+1,\rho}^* - w_{t,\rho}^*\|$$
$$\overset{(i)}{\leq}2\alpha\beta L_w\left\|\frac{F(x_t) - F(x_{t+1})}{\alpha} + \rho w_t\right\|\|\nabla F(x_t)w_t\| + \beta\rho\|w_t - w_{t,\rho}^*\|^2 + \frac{4}{\beta\rho}\|w_{t+1,\rho}^* - w_{t,\rho}^*\|^2$$
$$\leq 2\alpha\beta L_w M\left(\sqrt{K}M^2 + \frac{\|R(x_t)\|}{\alpha} + \rho\right) + \beta\rho\|w_t - w_{t,\rho}^*\|^2 + \frac{4\alpha^2 L_w^2 M^2}{\beta\rho}. \tag{38}$$

Then substituting eq. (36), eq. (37) and eq. (38) into eq. (35), we have

$$\|w_{t+1} - w_{t+1,\rho}^*\|^2 \leq (1 - \beta\rho)\|w_t - w_{t,\rho}^*\|^2 + 2\frac{\beta}{\alpha}\|R(x_t)\| + \beta^2(\rho + \sqrt{K}M^2 + \|R(x_t)\|)^2$$
$$\quad + \alpha^2 L_w^2 M + 2\alpha\beta L_w M\left(\sqrt{K}M^2 + \frac{\|R(x_t)\|}{\alpha} + \rho\right) + \frac{4\alpha^2 L_w^2 M^2}{\beta\rho}$$
$$\leq (1 - \beta\rho)\|w_t - w_{t,\rho}^*\|^2 + \alpha\beta K\ell(M+1)M^2 + \beta^2(C_1')^2$$
$$\quad + \alpha^2 L_w^2 M + 2\alpha\beta L_w MC_1' + \frac{4\alpha^2 L_w^2 M^2}{\beta\rho},$$

where the last inequality follows from Lemma 5 and $C_1' \geq \sqrt{K}M^2 + \rho + \frac{\alpha K\ell(M+1)M^2}{2}$. Then we do telescoping over $t = 0, 1, ..., T-1$

$$\|w_T - w_{T,\rho}^*\|^2 \leq (1 - \beta\rho)^T \|w_0 - w_{0,\rho}^*\|^2 + \frac{\alpha}{\rho}K\ell(M+1)M^2 + \frac{\beta}{\rho}(C_1')^2$$
$$+ \frac{\alpha^2}{\beta\rho}L_w^2 M + \frac{2\alpha L_w M}{\rho}C_1' + \frac{4\alpha^2 L_w^2 M^2}{\beta^2\rho^2}.$$

Then substituting the above inequality into eq. (34), we have

$$\|\nabla F(x_t)w_t - \nabla F(x_t)w_t^*\|$$
$$\leq \sqrt{K}M\Big[(1 - \beta\rho)^t \|w_0 - w_{0,\rho}^*\|^2 + \frac{\alpha}{\rho}K\ell(M+1)M^2 + \frac{\beta}{\rho}(C_1')^2$$
$$+ \frac{\alpha^2}{\beta\rho}L_w^2 M + \frac{2\alpha L_w M}{\rho}C_1' + \frac{4\alpha^2 L_w^2 M}{\beta^2\rho^2}\Big]^{\frac{1}{2}} + \sqrt{\rho}$$
$$= \mathcal{O}\Big((1 - \beta\rho)^{\frac{t}{2}}\|w_0 - w_{0,\rho}^*\| + \sqrt{\frac{\alpha}{\rho^2}} + \sqrt{\frac{\beta}{\rho}} + \frac{\alpha}{\beta\rho^2} + \sqrt{\rho}\Big).$$

Since we run projected gradient descent for the strongly convex function $J(w_n) = \frac{1}{2}\|\nabla F(x_0)w_n\|^2 + \frac{\rho}{2}\|w_n\|^2$ in the N-loop in Algorithm 4, according to Theorem 10.5 (Garrigos & Gower, 2023), we have

$$\|w_0 - w_{0,\rho}^*\|^2 = \|w_N - w_{0,\rho}^*\|^2 \leq 2\Big(1 - \frac{\rho}{M^2 + \rho}\Big)^N.$$

Thus, $\|w_0 - w_{0,\rho}^*\| = \mathcal{O}(\epsilon)$ as $N = \Omega(\rho^{-1})$. CA distance takes the order of $\epsilon$ in every iteration by choosing $\rho = \mathcal{O}(\epsilon^2)$, $\beta = \mathcal{O}(\epsilon^4)$, $\alpha = \mathcal{O}(\epsilon^9)$, and $N = \Omega(\epsilon^{-2})$. The proof is complete. $\square$

### E.4 FORMAL VERSION OF ITS PROOF OF THEOREM 6

Let $\alpha, \beta, \rho, T$ satisfy all requirements for Theorem 2 with $\delta < \frac{1}{2}$. Moreover, for $\rho = \mathcal{O}(\epsilon^2), N = \Omega(\epsilon^{-2}), \beta \leq \frac{\delta\rho\epsilon^2}{60(1+KM^4)} = \mathcal{O}(\epsilon^4), n_s \geq \max\{K\sigma^2, \frac{36K\sigma^2 M^2(6+20\beta\rho)}{\delta\rho^2\epsilon^2}\} = \Omega(\epsilon^{-6})$ and $\alpha \leq \sqrt{\frac{\delta\beta^2\rho^2\epsilon^2}{12L_w^2(2M^2+4K\sigma^2)(\beta\rho+1)}} = \mathcal{O}(\epsilon^9)$ and $T = \Theta(\epsilon^{-11})$, we have the following theorem:

**Theorem 12.** *If Assumptions 1, 2 and 3 hold, with the values of the parameters mentioned above, we have that for each $t \leq T$,*

$$\|\nabla F(x_t)w_t - \nabla F(x_t)w_t^*\| = \mathcal{O}(\epsilon),$$

*with the probability at least $1 - \delta$.*

*Proof.* When $\tau = T$ and $t < \tau$, according to the definition of CA distance, we have

$$\|\nabla F(x_t)w_t - \nabla F(x_t)w_t^*\|$$
$$= \|\nabla F(x_t)w_t - \nabla F(x_t)w_{t,\rho}^* + \nabla F(x_t)w_{t,\rho}^* - \nabla F(x_t)w_t^*\|$$
$$\overset{(i)}{\leq} \|\nabla F(x_t)w_t - \nabla F(x_t)w_{t,\rho}^*\| + \|\nabla F(x_t)w_{t,\rho}^* - \nabla F(x_t)w_t^*\|$$
$$\overset{(ii)}{\leq} \sqrt{K}M\|w_t - w_{t,\rho}^*\| + \sqrt{\rho}, \tag{39}$$

where $(i)$ follows from Cauchy-Schwartz inequality and $(ii)$ follows from $\|\nabla f_i(x_t)\| \leq M$ for any $i \in [K]$ and Lemma 4. We then show that for any $t \leq \tau$, we have that $\mathbb{E}[\|w_t - w_{t,\rho}^*\|^2 | \tau = T] \leq \frac{\delta}{2}\epsilon^2$ by induction.

**Base case:** Since we run projected gradient descent for the strongly convex function $J(w_n) = \frac{1}{2}\|\nabla F(x_0)w_n\|^2 + \frac{\rho}{2}\|w_n\|^2$ in the N-loop in Algorithm 1, according to Theorem 10.5 (Garrigos & Gower, 2023), we have by choosing $\beta' \in (0, \frac{1}{M^2}]$

$$\|w_0 - w_{0,\rho}^*\|^2 = \|w_N - w_{0,\rho}^*\|^2 \leq 2\Big(1 - \frac{\rho}{M^2}\Big)^N.$$

Thus, $\|w_0 - w_{0,\rho}^*\|^2 = \mathcal{O}(\frac{\delta}{2}\epsilon^2)$ as $N = \Omega(\rho^{-1})$.

**Induction:** Assume we have that $\mathbb{E}[\|w_t - w_{t,\rho}^*\|^2|\tau = T] \leq \frac{\delta}{2}\epsilon^2$, we will show that $\mathbb{E}[\|w_{t+1} - w_{t+1,\rho}^*\|^2|\tau = T] \leq \frac{\delta}{2}\epsilon^2$ holds for any $t < \tau$ in the following proof. We first divide $\|w_{t+1} - w_{t+1,\rho}^*\|^2$ into three parts:

$$\|w_{t+1} - w_{t+1,\rho}^*\|^2 = \|w_{t+1} - w_{t,\rho}^*\|^2 + \|w_{t+1,\rho}^* - w_{t,\rho}^*\|^2 - 2\langle w_{t+1} - w_{t,\rho}^*, w_{t+1,\rho}^* - w_{t,\rho}^*\rangle. \tag{40}$$

For the first term on the RHS in the above inequality, we have that

$$\|w_{t+1} - w_{t,\rho}^*\|^2$$
$$\overset{(i)}{\leq} \left\|w_t - \beta\Big(\nabla G_2(x_t)^\top \nabla G_3(x_t)w_t + \rho w_t\Big) - w_{t,\rho}^*\right\|^2$$
$$= \|w_t - w_{t,\rho}^*\|^2 - 2\beta\langle\nabla G_2(x_t)^\top \nabla G_2(x_t)w_t + \rho w_t, w_t - w_{t,\rho}^*\rangle$$
$$\quad + \beta^2\|\nabla G_2(x_t)^\top \nabla G_3(x_t)w_t + \rho w_t\|^2$$
$$\overset{(ii)}{\leq} (1 - 2\beta\rho)\|w_t - w_{t,\rho}^*\|^2$$
$$\quad + 2\beta\left\langle w_t - w_{t,\rho}^*, \varepsilon_{t,2}^\top\nabla F(x_t)w_t + \nabla F(x_t)^\top\varepsilon_{t,3}w_t - \varepsilon_{t,2}^\top\varepsilon_{t,3}w_t\right\rangle$$
$$\quad + \beta^2\|\rho w_t + \nabla F(x_t)^\top\nabla F(x_t)w_t - \varepsilon_{t,2}^\top\nabla F(x_t)w_t - \nabla F(x_t)^\top\varepsilon_{t,3}w_t + \varepsilon_{t,2}^\top\varepsilon_{t,3}w_t\|^2, \tag{41}$$

where $(i)$ follows from the non-expansiveness of projection and $(ii)$ follows from properties of strong convexity and Cauchy-Schwartz inequality. Taking the conditional expectation of equation 41, we have that for any $a_1 > 0$,

$$\mathbb{E}[\|w_{t+1} - w_{t,\rho}^*\|^2|\tau = T]$$
$$\leq \frac{\delta}{2}(1 - 2\beta\rho)\epsilon^2 + 2\beta\mathbb{E}[\|w_t - w_{t,\rho}^*\|\|\varepsilon_{t,2}^\top\nabla F(x_t)w_t + \nabla F(x_t)^\top\varepsilon_{t,3}w_t - \varepsilon_{t,2}^\top\varepsilon_{t,3}w_t\||\tau = T]$$
$$\quad + \mathbb{E}[\beta^2\|\rho w_t + \nabla F(x_t)^\top\nabla F(x_t)w_t - \varepsilon_{t,2}^\top\nabla F(x_t)w_t - \nabla F(x_t)^\top\varepsilon_{t,3}w_t + \varepsilon_{t,2}^\top\varepsilon_{t,3}w_t\|^2|\tau = T]$$
$$\leq \beta(\mathbb{E}[a_1\|w_t - w_{t,\rho}^*\|^2 + \|\varepsilon_{t,2}^\top\nabla F(x_t)w_t + \nabla F(x_t)^\top\varepsilon_{t,3}w_t - \varepsilon_{t,2}^\top\varepsilon_{t,3}w_t\|^2/a_1|\tau = T]])$$
$$\quad + \frac{\delta}{2}(1 - 2\beta\rho)\epsilon^2 + 5\beta^2\rho^2 + 5\beta^2 KM^4 + 5\beta^2\mathbb{E}[M^2\|\epsilon_{t,2}\|^2|\tau = T]$$
$$\quad + 5\beta^2\mathbb{E}[M^2\|\epsilon_{t,3}\|^2|\tau = T] + 5\beta^2\mathbb{E}[\|\epsilon_{t,2}\|^2\|\epsilon_{t,3}\|^2|\tau = T], \tag{42}$$

where the last inequality is due to that for $t \leq \tau = T$, and for any $i \in [K]$, we have that $\|\nabla f_i(x_t)\| \leq M$. Then for the second term on the RHS in eq. (40), we have

$$\mathbb{E}[\|w_{t+1,\rho}^* - w_{t,\rho}^*\|^2|\tau = T] \leq \mathbb{E}[L_w^2\|x_t - x_{t+1}\|^2|\tau = T]$$
$$= \mathbb{E}[L_w^2\alpha^2\|\nabla F(x_t, s_{t,1})w_t\|^2|\tau = T]$$
$$\leq \mathbb{E}[\alpha^2 L_w^2(M + \|\epsilon_{t,1}w_t\|)^2|\tau = T], \tag{43}$$

where the first inequality is due to Lemma 3, where $L_w = \mathcal{O}(\rho^{-1})$. Then for the last term on the RHS in eq. (40), for any $a_2 > 0, a_3 > 0$, we have that

$$\mathbb{E}[-2\langle w_{t+1} - w_{t,\rho}^*, w_{t+1,\rho}^* - w_{t,\rho}^*\rangle|\tau = T]$$
$$\leq \mathbb{E}[2\|w_{t+1} - w_{t,\rho}^*\|\|w_{t+1,\rho}^* - w_{t,\rho}^*\||\tau = T]$$
$$\leq \mathbb{E}[2(\|w_{t+1} - w_t\| + \|w_t - w_{t,\rho}^*\|)\|w_{t+1,\rho}^* - w_{t,\rho}^*\||\tau = T]$$
$$\leq \mathbb{E}\left[a_2\|w_{t+1} - w_t\|^2 + \frac{1}{a_2}\|w_{t+1,\rho}^* - w_{t,\rho}^*\|^2 + a_3\|w_t - w_{t,\rho}^*\|^2 + \frac{1}{a_3}\|w_{t+1,\rho}^* - w_{t,\rho}^*\|^2|\tau = T\right]$$
$$\overset{(i)}{\leq} \mathbb{E}\left[a_2\beta^2\|\nabla G_2(x_t)^\top\nabla G_3(x_t)w_t + \rho w_t\|^2 + a_3\frac{\delta}{2}\epsilon^2 + \left(\frac{1}{a_2} + \frac{1}{a_3}\right)\alpha^2 L_w^2(M + \|\epsilon_{t,1}w_t\|)^2|\tau = T\right]$$
$$\leq a_2(5\beta^2\rho^2 + 5\beta^2 KM^4 + 5\beta^2\mathbb{E}[M^2\|\epsilon_{t,2}\|^2|\tau = T]$$
$$\quad + 5\beta^2\mathbb{E}[M^2\|\epsilon_{t,3}\|^2|\tau = T] + 5\beta^2\mathbb{E}[\|\epsilon_{t,2}\|^2\|\epsilon_{t,3}\|^2|\tau = T])$$
$$\quad + \mathbb{E}\left[\left(\frac{1}{a_2} + \frac{1}{a_3}\right)\alpha^2 L_w^2(M + \|\epsilon_{t,1}w_t\|)^2|\tau = T\right] + a_3\frac{\delta}{2}\epsilon^2, \tag{44}$$

where $(i)$ follows from the non-expansiveness of projection and equation 43, and the last inequality is from equation 42. Then substituting eq. (42), eq. (43) and eq. (44) into eq. (40), we have

$$\mathbb{E}[\|w_{t+1} - w^*_{t+1,\rho}\|^2 | \tau = T]$$

$$\leq (1 - 2\beta\rho + \beta a_1 + a_3)\frac{\delta}{2}\epsilon^2$$

$$+ \beta^2(5\rho^2 + 5KM^4)(1 + a_2)$$

$$+ M^2\left(\frac{3\beta}{a_1} + 5\beta^2 + 5\beta^2 a_2\right)\mathbb{E}[\|\varepsilon_{t,2}\|^2 | \tau = T]$$

$$+ M^2\left(\frac{3\beta}{a_1} + 5\beta^2 + 5\beta^2 a_2\right)\mathbb{E}[\|\varepsilon_{t,3}\|^2 | \tau = T]$$

$$+ \left(\frac{3\beta}{a_1} + 5\beta^2 + 5\beta^2 a_2\right)\mathbb{E}[\|\varepsilon_{t,2}\|^2\|\varepsilon_{t,3}\|^2 | \tau = T]$$

$$+ \mathbb{E}\left[\left(1 + \frac{1}{a_2} + \frac{1}{a_3}\right)\alpha^2 L_w^2(M + \|\epsilon_{t,1} w_t\|)^2 | \tau = T\right]$$

$$\leq (1 - 2\beta\rho + \beta a_1 + a_3)\frac{\delta}{2}\epsilon^2$$

$$+ \beta^2(5\rho^2 + 5KM^4)(1 + a_2)$$

$$+ M^2\left(\frac{3\beta}{a_1} + 5\beta^2 + 5\beta^2 a_2\right)\left(\frac{4K\sigma^2}{n_s} + \frac{2K^2\sigma^4}{n_s^2}\right)$$

$$+ \left(1 + \frac{1}{a_2} + \frac{1}{a_3}\right)\alpha^2 L_w^2\left(2M^2 + \frac{4K\sigma^2}{n_s}\right), \tag{45}$$

where the last inequality is due to that for any $i \in [3]$,

$$\mathbb{E}[\|\epsilon_{t,i}\| | \tau = T] \leq \sqrt{\mathbb{E}[\|\epsilon_{t,i}\|^2 | \tau = T]} \leq \sqrt{\mathbb{E}[\|\epsilon_{t,i}\|^2]/\mathbb{P}(\tau = T)} \leq \sqrt{\frac{2K}{n_s}}\sigma$$

and

$$\mathbb{E}[\|\epsilon_{t,2}\|\|\epsilon_{t,3}\| | \tau = T] \leq \sqrt{\mathbb{E}[\|\epsilon_{t,2}\|^2\|\epsilon_{t,3}\|^2 | \tau = T]}$$

$$\leq \sqrt{\mathbb{E}[\|\epsilon_{t,2}\|^2\|\epsilon_{t,3}\|^2]/\mathbb{P}(\tau = T)}$$

$$\leq \sqrt{\mathbb{E}[\|\epsilon_{t,2}\|^2]\mathbb{E}[\|\epsilon_{t,3}\|^2]/\mathbb{P}(\tau = T)}$$

$$\leq \frac{\sqrt{2}K\sigma^2}{n_s}.$$

According to equation 45, with $a_1 = 0.5\rho, a_2 = 1, a_3 = 0.5\beta\rho, \beta \leq \frac{\delta\rho\epsilon^2}{60(1+KM^4)}, n_s \geq \max\{K\sigma^2, \frac{36K\sigma^2 M^2(6+20\beta\rho)}{\delta\rho^2\epsilon^2}\}$ and $\alpha \leq \sqrt{\frac{\delta\beta^2\rho^2\epsilon^2}{12L_w^2(2M^2+4K\sigma^2)(\beta\rho+1)}}$, we have that

$$\mathbb{E}[\|w_{t+1} - w^*_{t+1,\rho}\|^2 | \tau = T] \leq \frac{\delta}{2}\epsilon^2.$$

We then complete our induction and prove that for any $t < \tau$, we have that $\mathbb{E}[\|w_{t+1} - w^*_{t+1,\rho}\|^2 | \tau = T] \leq \frac{\delta}{2}\epsilon^2$.

As a result, we have that

$$\mathbb{P}\left(\|w_{t+1} - w^*_{t+1,\rho}\|^2 > \epsilon^2 \Big| \tau = T\right) \leq \frac{\mathbb{E}\left[\|w_{t+1} - w^*_{t+1,\rho}\|^2 \Big| \tau = T\right]}{\epsilon^2} \leq \frac{\delta}{2},$$

where the first probability is due to Markov inequality. Thus we have that

$$\mathbb{P}\left(\|w_{t+1} - w^*_{t+1,\rho}\|^2 \leq \epsilon^2\right)$$

$$\geq 1 - \mathbb{P}(\tau < T) - \mathbb{P}\left(\|w_{t+1} - w^*_{t+1,\rho}\|^2 \Big| \tau = T\right)\mathbb{P}(\tau = T)$$

$$\geq 1 - \delta, \tag{46}$$

where the last inequality is because our parameters satisfy all the requirements in Theorem 2, thus $\mathbb{P}(\tau < T) \leq \frac{\delta}{2}$. Then based on equation 39, by setting $\rho = \mathcal{O}(\epsilon^2)$, we have that $\|\nabla F(x_t)w_t - \nabla F(x_t)w_t^*\| = \mathcal{O}(\epsilon)$ with probability at least $1 - \delta$ for each iteration $t$, which completes the proof. $\qquad\square$

