# OpenReview forum: "MGDA Converges under Generalized Smoothness, Provably"
_ICLR.cc/2025/Conference — ICLR 2025 Poster_

### Official Review · Reviewer_moXU · 2024-10-29

**Soundness:** 2
**Presentation:** 3
**Contribution:** 2
**Rating:** 5
**Confidence:** 3

**Summary:**

This paper investigates a broader class of generalized $\ell$-smooth loss functions, where $\ell$ is a non-decreasing function of the gradient norm. We analyze the multiple gradient descent algorithm (MGDA) and its stochastic variant for addressing generalized ℓ\ellℓ-smooth multi-objective optimization (MOO) problems. Its comprehensive convergence analysis demonstrates that these algorithms converge to an $\epsilon$-accurate Pareto stationary point, ensuring guaranteed average conflict-avoidant distances.

**Strengths:**

1.Writing: This work is presented with good writing style, where the summarized problems with detailed explanations make it easy for readers to understand the problem addressed in this article.

2.Novelty: The paper investigates a more general and realistic class of generalized $\ell$-smooth loss functions, which has been rarely considered in previous work. Additionally, a warm start strategy is proposed in the algorithm design to initialize the parameter $\omega$ for enhanced performance.

3.Experiments: Several scenarios and recent baselines are considered, implying improvements in accuracy and robustness under various distributional shifts.

**Weaknesses:**

1.Unreliable theoretical analysis: The theoretical work in this paper appears to lack reliability. A simple review of the proofs in the appendix reveals that the proof of Corollary 1 is entirely incorrect.

The inequality
$$
\left\|\nabla F\left(x_t\right) w_t-\nabla F\left(x_t\right) w_t^*\right\|^2 \leq\left\|\nabla F\left(x_t\right) w_t\right\|^2-\left\|\nabla F\left(x_t\right) w_t^*\right\|^2,
$$
does not hold.

2.Originality of the algorithm. Could you clarify the differences between the algorithm presented in this paper and MGDA [1]?  Could you elaborate on how the algorithm mentioned in this paper improves the performance of MGDA?

[1]Jean-Antoine Désidéri. Multiple-gradient descent algorithm (mgda) for multiobjective optimization.  Comptes Rendus Mathematique, 350(5-6):313–318, 2012.

3.Warm start. Can the warm start strategy be applied to other multi-objective optimization algorithms? Would incorporating this strategy significantly improve algorithm performance?

4.Computational complexity.
The paper may not provide a comprehensive assessment of the computational efficiency and practicality of the proposed method in real-world applications. Like the computational complexity analysis or empirical time/memory cost.

**Questions:**

For specific questions, please refer to the "Weakness" section.
I would be willing to raise my score if these  questions concerned are well addressed.

---

> ### Author Response · Authors · 2024-11-21
>
> We thank the reviewer moXU for the time and valuable feedback! Here are our responses.
>
> **Weakness 1:** We believe our proof is reliable and kindly request the reviewer to rethink the point. This inequality holds and here are the details steps. It is a widely used inequality in the analysis of Multi-objective optimization, such as Lemma 6 in MoCo [a].
> $$\\|\nabla F(x_t)w_t-\nabla F(x_t)w_t^*\\|^2
> =\\|\nabla F(x_t)w_t\\|^2+\\|\nabla F(x_t)w_t^*\\|^2-2\langle\nabla F(x_t)w_t,\nabla F(x
> _t)w_t^*\rangle$$
> $$\leq\\|\nabla F(x_t)w_t\\|^2+\\|\nabla F(x_t)w_t^*\\|^2-2\\|\nabla F(x_t)w_t^*\\|^2$$
> $$=\\|\nabla F(x_t)w_t\\|^2-\\|\nabla F(x_t)w_t^*\\|^2$$
> $$\leq\\|\nabla F(x_t)w_t\\|^2$$
>
> where the first inequality holds because of the optimality condition of $w_t^*=\arg\min_{w\in\mathcal{W}} \frac{1}{2}\\|\nabla F(x_t)w\\|^2$ that for any $w\in\mathcal{W}$, $$\langle w, \nabla F(x_t)^\top\nabla F(x_t)w_t^*\rangle\geq\langle w_t^*, \nabla F(x_t)^\top\nabla F(x_t)w_t^*\rangle=\\|\nabla F(x_t)w_t^*\\|^2.$$
> We have included it in the revision to make it clearer.
>
> [a] Fernando, H., Shen, H., Liu, M., Chaudhury, S., Murugesan, K., & Chen, T. (2022). Mitigating gradient bias in multi-objective learning: A provably convergent stochastic approach. arXiv preprint arXiv:2210.12624.
>
> **Weakness 2:** Thank you for the question. We believe there may be a misunderstanding regarding the focus of this paper. Our work investigates the convergence of MGDA under generalized smoothness rather than proposing new algorithms, and we would like to clarify our contributions accordingly.
>
> From a theoretical standpoint, this paper is the first to examine the convergence of MGDA under generalized smoothness conditions, which has not been previously addressed. All prior MGDA-related studies rely on standard $L$-smooth and bounded-gradient assumptions, which may not hold in neural network training.  As a result, we could not apply the analytical techniques in those papers but start new analysis.
>
> From the algorithmic standpoint, the concept of warm start is essential in our algorithm to ensure an iteration-wise CA distance under generalized smoothness as explained in lines 274-280. On the contrary, MGDA in [1] can only have a convergence guarantee under a deterministic setting with stand $L$-smooth assumption. Last but not least, we provide a guarantee of the fast approximation (FA), which saves the computational cost in MGDA. In general, our paper studies a broad version of MGDA within an unexplored setting, generalized smoothness. Thus, it is not fair to only compare the algorithm structure. We have clearly demonstrated our focus in the abstract.
>
> **Weakness 3:** Yes, the warm start strategy can be applied to other multi-objective optimization algorithms. For MGDA-based methods, both MoCo [a] and MoDo [b] can incorporate the warm start as an add-on. However, the MoDo-warm start will exactly recover to our algorithm. Therefore, we evaluate the performance of the MoCo-warm start on the Cityscapes and NYU-v2 datasets. As shown in the following tables, the warm start strategy improves performance. This result has been added in Section B.3.
> |Method|Segmentation||Depth||$\Delta m\% \downarrow$|
> |-|-|-|-|-|-|
> ||mIoU $\uparrow$|Pix Acc $\uparrow$| Abs Err $\downarrow$|Rel Err $\downarrow$||
> |MoCo| 75.42| 93.55|0.0149|34.19|9.90|
> |MoCo-warm start|75.48| 93.54|0.0148| 31.43| 7.32|
>
> *Table: Multi-task supervised learning on Cityscapes dataset.*
>
> | Method| Segmentation||Depth|||Surface Normal|||| $\Delta m\% \downarrow$|
> |-|-|-|-|-|-|-|-|-|-|-|
> || mIoU $\uparrow$| Pix Acc $\uparrow$ | Abs Err $\downarrow$ | Rel Err $\downarrow$ | Angle Distance $\downarrow$ | Mean| Median| Within $11.25^\circ$ $\uparrow$ | Within $22.5^\circ$ $\uparrow$ | Within $30^\circ$ $\uparrow$ |
> | MoCo| 40.30| 66.07| 0.5575| 0.2135| 26.67| 21.83| 25.61| 51.78| 64.85| 0.16|
> | MoCo-warm start| 38.40| 64.40| 0.5377| 0.2315| 26.04| 20.57| 27.11| 54.02| 66.63| -0.88|
>
> *Table: Multi-task supervised learning on NYU-v2 dataset.*
>
> [b] Chen, L., Fernando, H., Ying, Y., & Chen, T. (2024). Three-way trade-off in multi-objective learning: Optimization, generalization and conflict-avoidance. Advances in Neural Information Processing Systems, 36.
>
> **Weakness 4:** For computational complexity, we compare it in terms of the number of gradient computations. Methods can be broadly categorized into MGDA-based and MTL-based approaches. Assuming $T$ epochs and $K$ objectives (tasks), MGDA-based methods—including MGDA, PCGrad, CAGrad, GradDrop, MoCo, MoDo, and Nash-MTL—require
> $\mathcal{O}(KT)$ gradient computations. In contrast, MTL-based methods such as LS, SI, RLW, DWA, UW, and FAMO require only $\mathcal{O}(T)$ gradient computations. In most cases, MGDA-based methods demonstrate superior performance due to their ability to mitigate gradient conflicts. We will clarify this distinction further in the revised paper.

---

> > ### Comment · Reviewer_moXU · 2024-11-22
> >
> > Thank the authors for the response. The response and the revision has addressed the most of my concerns. As an result, I increase my rating to 5.

---

> > > ### Author Response · Authors · 2024-11-22
> > >
> > > Dear Reviewer moXU,
> > >
> > > Thanks so much for your updates and for raising your score! We are happy that our responses address your concerns.
> > >
> > > Best, Authors

---

### Official Review · Reviewer_Ksvn · 2024-11-01

**Soundness:** 2
**Presentation:** 3
**Contribution:** 2
**Rating:** 5
**Confidence:** 2

**Summary:**

This paper offers a rigorous convergence analysis for both deterministic and stochastic variants of the Multiple Gradient Descent Algorithm (MGDA) under a generalized $\ell$-smoothness assumption. The authors demonstrate that these methods converge to an $\epsilon$-accurate Pareto stationary point while maintaining an $\epsilon$-level average conflict-avoidant (CA) distance over all iterations. The sample complexity is shown to be $\mathcal{O}(\epsilon^{-2})$ in the deterministic setting and $\mathcal{O}(\epsilon^{-4})$ in the stochastic setting. Additionally, the authors introduce a warm start strategy to enable more precise control over iteration-wise CA distance. They also analyze MGDA-FA, a variant that reduces the computational complexity to $\mathcal{O}(1)$ in both time and space, making it notably efficient.

**Strengths:**

1. The theoretical analysis is comprehensive and robust.
2. The paper is clearly written and well-organized, making the concepts easy to follow.

**Weaknesses:**

1. Assumption 2, which posits that $\phi(a) = \frac{a^2}{2\ell(2a)}$ is monotonically increasing, restricts $\ell(a) \leq \mathcal{O}(a^2)$. This places a strict limitation on the class of generalized $\ell$-smooth functions considered.
2. The paper's novelty is somewhat limited. From an algorithmic standpoint, MGDA was introduced several years ago, and the fast approximation presented here constitutes a relatively minor modification.
3. The novelty of the analytical techniques is also limited, as the approach used closely resembles that of [1].
4. The experimental results would benefit from the addition of confidence intervals to clarify statistical significance, particularly given the small differences in performance observed.
5. In the main theorems, the authors use big-O notation without specifying constants. Providing explicit constants would enhance clarity and interpretability.

**Reference**:

[1] Li, Haochuan, et al. "Convex and non-convex optimization under generalized smoothness." Advances in Neural Information Processing Systems 36 (2024).

**Questions:**

1. In lines 389 and 393, why are $i \in [3]$ and $j \in [3]$ specified? What is the rationale behind selecting only the first three sample collections?
2. In Section 5.2, the analysis involves mini-batch algorithms, yet the effect of batch size on the results is not discussed. How would adjusting the batch size impact the findings?

---

> ### Author Response · Authors · 2024-11-21
>
> We thank the reviewer Ksvn for the time and valuable feedback! Here are our responses.
>
> **Weakness 1:** Thank you for the question. This assumption is already the weakest one in the generalized smooth studies, where most of them assume $\ell(a)$ is a linear function of $a$. Even for the single task problem, only the convergence with high probability can be obtained in the stochastic setting when $\ell(a)=\mathcal O(a^2)$ and no convergence is guaranteed even for deterministic setting when $\ell(a)>\mathcal O(a^2)$. The relaxation of this limitation is a very interesting topic and deserves more attention from both single-objective and multi-objective societies.
>
> **Weakness 2:** Thank you for the question. We believe there may be a misunderstanding regarding the focus of this paper. Our work investigates the convergence of MGDA under generalized smoothness rather than proposing new algorithms, and we would like to clarify our contributions accordingly.
>
> From a theoretical standpoint, this paper is the first to examine the convergence of MGDA under generalized smoothness conditions, which has not been previously addressed. All prior MGDA-related studies rely on standard $L$-smooth and bounded-gradient assumptions, which may not hold in neural network training.  Because of that, we could not apply the analytical techniques in those papers but start the analysis from the generalized smoothness.
>
> From the algorithmic standpoint, fast approximation implementation is not our main contribution, either. A similar idea was proposed by FAMO to significantly save computational cost as we mentioned in lines 319-322. However, there is no theoretical convergence analysis of this algorithm. In this paper, we not only fill this gap but also relax the assumption to generalized smoothness. Finally, the concept of warm start is essential in our algorithm to ensure an iteration-wise CA distance under generalized smoothness, representing a key aspect of our algorithmic innovation. Once again, our paper primarily emphasizes the convergence analysis of MGDA under generalized smoothness, as our novel and significant contribution.
>
> **Weakness 3:** Thanks for the question. We would like to clarify our analytical techniques accordingly. [1] only focuses on the single-task optimization problem and it takes non-trivial efforts to get our theoretical results in a multi-objective setting.  [1] requires the objective functions to be generalized smooth. However, in this paper, any linear combinations of generalized smooth functions are no longer generalized smooth. In a stochastic setting, it is easy to show the single function value bounded with high probability [1]. However, in the MOO setting it requires more effort to control function values for every task. Moreover, [1] does not need to find the CA direction and all exiting methods on MOO to get the CA directions do not work due to the unbounded gradient introduced by generalized smoothness.
> In the stochastic setting, bounding the single-function value with high probability is relatively straightforward [1]. However, in the MOO setting, it requires more effort to control the function values across all tasks.
>
> Furthermore, [1] does not address the problem of finding the conflict-avoidant (CA) direction, which is essential in MOO. Existing MOO methods for determining CA directions fail under generalized smoothness due to the unbounded gradients it introduces. This is the first work to solve the challenges of finding CA direction under generalized smoothness.
>
> Importantly, we incorporate the warm start procedure as both an analytical and algorithmic novelty. To the best of our knowledge, no prior single-loop MGDA-based algorithm has achieved iteration-wise CA distance, not to mention the generalized smoothness settings. By leveraging the warm start, we can control the error term of the weight initialization from the beginning, ensuring improved control over iteration-wise CA distance.
>
> Finally, while the fast approximation is more efficient, it lacks theoretical guarantees in prior work. We address this gap by providing a convergence analysis along with a CA distance bound. In a word, these contributions establish the novelty and significance of our analytical techniques.
>
> **Weakness 4:** Thank you for the suggestion. To address your concern, in the revised pdf,  we report the mean and standard deviation of $\Delta m\%$ since it serves as an overall metric to evaluate the performance. For our method, $\Delta m\%=3.94\pm1.19$ can be found in the Table 3.

---

> ### Author Response · Authors · 2024-11-21
>
> **Weakness 5:** Thanks for pointing it out. In the deterministic setting, the gap between the initial function and optimal value  $\Delta$ is bounded and can be assumed as a constant since the loss function is typically non-negative.  Thus $M=\mathcal O(\Delta)$ is also a constant. For the stochastic setting,  $M=\mathcal O(\frac{\Delta}{\delta})$ and we provide the dependencies on other parameters:  $\rho\le \mathcal O(\delta^2 \epsilon^2), \beta\le \mathcal O(\min\\{\frac{\delta\epsilon^2}{M^4},\frac{\rho}{M^2}\\}), \alpha\le \mathcal O(\min\\{\beta, \frac{\rho}{\ell(M+1)^2},  \frac{\rho}{M^2}, \frac{\delta }{\rho T},\frac{1}{\beta TM^2}\\})$ and $T\ge\Theta(\max\\{\frac{1}{\delta \alpha \epsilon^2},\frac{M^2}{\delta^2 \epsilon^4}\\})$.
>
> **Question 1:** We have used 3 different samples in Algorithm 3. The first sample comes from the model update and a double-sampling strategy is used in the weight update process. Therefore, we specify $i\in[3]$. We have made it clearer in the revision.
>
> **Question 2:** Thanks for pointing out this problem. To get the average CA distance bounded, there are no requirements on $n_s$.
> In our theorem 6 for iteration-wise CA distance, we require $n_s\ge \max\\{\frac{1}{K\sigma},\frac{36K\sigma(6+20\beta\rho)}{\delta\rho^2\epsilon^2}\\}$ to bound $\mathbb E[\\|w_{t}-w_{t,\rho}^*\\|^2$ in the order of $\mathcal O(\delta \epsilon^2)$ for any $t\le T$. A larger $n_s$ makes the gap smaller but increases the computation. However, a smaller batch will fail to guarantee the iteration-wise CA bounded with high probability.

---

> > ### Author Response · Authors · 2024-11-24
> >
> > Dear Reviewer Ksvn,
> > As the discussion period is approaching its end, we have yet to receive your feedback on our responses. We would greatly appreciate it if you could confirm whether we have addressed your concerns.
> >
> > We would be happy to clarify or provide additional information if needed. Additionally, we would be grateful if you could reconsider the score based on our clarifications.
> >
> > Thank you for your time and attention.
> >
> > Best regards,
> > The authors

---

### Official Review · Reviewer_YM5v · 2024-11-03

**Soundness:** 3
**Presentation:** 2
**Contribution:** 2
**Rating:** 6
**Confidence:** 2

**Summary:**

This paper studies the multi-objective optimization (MOO) problem under generalized $ell$-smoothness, where $\ell$ is a general non-decreasing function of gradient norm. The authors provide the first convergence analysis of multiple gradient descent algorithm (MGDA) (and its variant with fast approximation) under this setting, in terms of $\epsilon$-Pareto stationarity and $\epsilon$-level conflict-avoidant (CA) distance. The resulting complexities with respect to $\epsilon$-Pareto stationarity and $\epsilon$-level average CA distance match the best known results.

**Strengths:**

- The analysis of MGDA under generalized smoothness is new. Their analysis further relax the assumptions such as bounded function values or bounded gradients assumed in prior work.
- The authors use a general notion of generalized smoothness, i.e., in terms of a non-decreasing function $\ell$ instead of the original one with $ell(a) = L_0 + L_1 a$.
- The authors further study a variant of MGDA, which approximates the gradient in each iteration to save memory and time.

**Weaknesses:**

- The notation of this paper is a bit unclear. In multiple places, the authors introduce something before they define the notation, so the write-up should be refined. For example, the authors should define $\mathcal{W}$ in Definition 3.
- The preliminaries on generalized smoothness from Section 2.1 is redundant, as those are not proposed by this work and not the main contribution here.
- All the complexity results of this paper are only stated in terms of $\epsilon$. A more clear statements with dependencies on other problem parameters would be better.

**Questions:**

- What is the meaning of minimizing a vector as in Eq. (1)? I suggest that the authors may make it clearer, in the context of multi-objectives.
- Could the authors elaborate how they computed the local smoothness for Figure 1?
- What is $f_*$ in the definition of $\Delta$ in Line 357?
- Take Theorem 1 as an example. Could the authors provide a more detailed discussion on the constants $c$, $F$ and $M$, like how tight they can be in some special case? I found the current convergence result is hard to parse.
- Would generalized smoothness bring any benefits in practice, like providing guidance on how to choose the step sizes?

---

> ### Author Response · Authors · 2024-11-21
>
> We thank the reviewer YM5v for the time and valuable feedback! Here are our responses.
>
> **Weakness 1:** Thanks for pointing it out. We have moved the definition of $\mathcal{W}$ from section 2.3 to Definition 3 in the revision. We also add a notation summary in the Appendix in the revision.
>
> **Weakness 2:** Though the preliminaries in Section 2 are not our original contribution, we believe they are essential for providing readers who may be less familiar with this topic a foundational understanding. Including these concepts ensures clarity and a good presentation of our paper. However, we would like to shorten it slightly without hurting its readability.
>
> **Weakness 3:** Thanks for pointing it out. In the deterministic setting, $\Delta$ is the gap between the initial function value and optimal value, which is bounded and can be assumed as a constant since the loss function is typically non-negative.  Thus $M=\mathcal O(\Delta)$ is also a constant. For the stochastic setting,  $M$ depends on $\frac{1}{\delta}$ and we provide the dependencies on other parameters:  $\rho\le \mathcal O(\delta^2 \epsilon^2), \beta\le \mathcal O(\min\\{\frac{\delta\epsilon^2}{M^4},\frac{\rho}{M^2}\\}), \alpha\le \mathcal O(\min\\{\beta, \frac{\rho}{\ell(M+1)^2},  \frac{\rho}{M^2}, \frac{\delta }{\rho T},\frac{1}{\beta TM^2}\\})$ and $T\ge\Theta(\max\\{\frac{1}{\delta \alpha \epsilon^2},\frac{M^2}{\delta^2 \epsilon^4}\\})$.
>
> **Question 1:** Minimizing a vector as in Eq. (1) means minimizing different objective functions simultaneously. This is a standard formulation in the context of multi-objective optimization as shown in MoCo, SDMGrad, FairGrad, etc.
>
> **Question 2:** We have mentioned in lines 512-513 that the local smoothness is computed according to the method in Section H.3 in the paper, "Why gradient clipping accelerates training: A theoretical justification for adaptivity" with available code in https://github.com/JingzhaoZhang/why-clipping-accelerates. The local smoothness is computed as
> $$\widehat{L}(x_t)=\max_{\gamma\in\{\delta,2\delta,...,1\}}\frac{\\|\nabla f(x_t+\gamma d)-\nabla f(x_t)\\|}{\\|\gamma d\\|},$$
> where $\delta\in(0,1)$ is a small value, $d=x_{t+1}-x_t$.
>
> **Question 3:** Thanks for pointing it out. This is a typo here that $f_*$ should be $f_i^*$. We have fixed it in the revision.
>
> **Question 4:** Thanks for pointing it out. For Theorem 1, $c_1,c_2,c_3$ are any fixed constant, we can set all of them to $1$. Then $F=\Delta+3$.  In the deterministic setting, $\Delta$ is the gap between the initial function value and optimal value, which is bounded and can be assumed as a constant since the loss function is typically non-negative. If $\ell$ is the widely studied linear function,
> then we have $M=\mathcal O(\Delta)$ and [a] shows a total sample complexity of $\mathcal{O}(M^2\epsilon^{-2})$.
> Theorem 1 then requires $\alpha, \beta \le \mathcal O(M^{-2})$, thus it
> require $\mathcal{O}(M^3\epsilon^{-2})$. The additional complexity in our methods is introduced because we required smaller stepsize $\beta$ to update the CA direction which the CA direction is not needed in a single-task setting. However, in a deterministic setting, $M$ is a constant. Thus it should not be an issue.
>
> **Question 5:** Yes, the generalized smoothness condition brings benefits in practice. Under the generalized smoothness condition, we design more complicated step sizes, which are smaller compared with methods for standard $L$-smoothness in order to bound the function values for all tasks. Otherwise, the training of MGDA with larger stepsizes may diverge. The relaxation of requirements on step sizes needs additional operations such as clipping and momentum, which changes the structure of MGDA and is not the focus of this paper.

---

> > ### Author Response · Authors · 2024-11-24
> >
> > Dear Reviewer YM5v,
> > As the discussion period is approaching its end, we have yet to receive your feedback on our responses. We would greatly appreciate it if you could confirm whether we have addressed your concerns.
> >
> > We would be happy to clarify or provide additional information if needed. Additionally, we would be grateful if you could reconsider the score based on our clarifications.
> >
> > Thank you for your time and attention.
> >
> > Best regards,
> > The authors

---

> > > ### Comment · Reviewer_YM5v · 2024-11-25
> > >
> > > I thank the reviewer's response. For my Q1, my concern was that the authors did not make eq. (1) and its following discussion rigorous. The authors should have at least mentioned Pareto stationarity after eq. (1) to explicitly show what is the meaning of minimizing a vector of different function values. All the related works the authors mentioned carried out the introduction in that way. For Q4, if $c_1, c_2, c_3$ can be any fixed constants, could we just set them to be 1 and simplify the presentation of the main body? I am ok with other parts, and I slightly increase my score.

---

> > > > ### Author Response · Authors · 2024-11-25
> > > >
> > > > Dear Reviewer YM5v,
> > > >
> > > > Thank you for your valuable feedback and insights, which have significantly contributed to improving the quality of this paper. Based on your suggestions, we have made the following revisions.
> > > >
> > > >  For Q1, we have added the following discussions on Pareto stationarity after eq. (1).
> > > >
> > > > In MOO setting, we are interested in optimizing all the objectives simultaneously. However, this problem is challenging due to the gradient conflict that some objectives with larger gradients dominate the update direction at the sacrifice of
> > > > significant performance degeneration on the less-fortune objectives with smaller gradients. Thus, a  widely-adopted target is to find the Pareto stationary point $x$ that the performance of all objectives cannot be further improved without compromising some objectives.
> > > >
> > > >
> > > >
> > > >  For Q4, It is OK to set them to $1$ and in the revision, we set $c_1,c_2,c_3$ to $1$ to simplify the presentation of the main body.
> > > >
> > > >  Thanks so much for your suggestions and please let us know if you have further questions.
> > > >
> > > > The authors

---

### Official Review · Reviewer_Wanu · 2024-11-09

**Soundness:** 3
**Presentation:** 3
**Contribution:** 3
**Rating:** 8
**Confidence:** 2

**Summary:**

The authors analyze the problem of multi objective optimization (MOO),
$$\begin{align*}
F^\star = \min_{x\in \mathbb{R}^d} F(x) = (f_1(x), f_2(x), \ldots f_K(x))
\end{align*}
$$
where each $f_i:\mathbb{R}^m \to \mathbb{R}$ when each $f_i$ has generalized $\ell$-smoothness. Generalized $\ell$-smoothness implies that $\|\nabla^2 f_i(x)\| \leq \ell(\|\nabla f_i(x)\|), \forall x$ for a continuous non-decreasing positive function $\ell$. This subsumes $(L_0, L_1)$-smoothness, where $\ell(a) = L_0 + L_1 a$, (Zhang et al 2019) and standard $L$-smoothness, where $\ell(a) = L$. Further,  NNs do not satisfy stronger notions of $L$-smoothness but satisfy generalized smoothness.

The goal of MOO is to obtain $\epsilon$-accurate Pareto stationary point, defined as $\min_{w\in \mathcal{W}} \|\nabla F(x) w\|^2 \leq \epsilon^2$. This is referred to as $\epsilon$-CA distance (Conflict avoidant).

First, the authors analyze the existing MGDA algorithm in Algorithm 1 (Desideri 2012) and its stochastic variant in Algorithm 3, to obtain $\epsilon$-CA distance on average. The required sample complexity for these cases is $\mathcal{O}((\alpha\epsilon^2)^{-1}, (\beta\epsilon^2)^{-1})$(Theorem 1)  and $\mathcal{O}((\alpha \epsilon^2)^{-1} + \epsilon^{-4})$ (Theorem 2) respectively, which matches the sample complexity for single objective optimization under $L$-smoothness. Here, $\alpha$ and $\beta$ are step sizes to update  $w$ and $x$ in their algorithms.

Second, the authors consider a stronger metric, the per-iteration $\epsilon$-level CA distance, defined as $\|\nabla F(x_t)w_t - \nabla F(x_t) w_t^\star\| \leq O(\epsilon)$, where $x_t, w_t$ are iterates of the algorithm, and $w_t^\star \in \arg\min_{w\in \mathcal{W}} \|\nabla F(x_t) w\|^2$. To achieve this metric, the authors propose using a  warm-start for $w$ in deterministic setting (Algorithm 1), and warmstart with increasing batch size for stochastic setting (Algorithm 3), yielding sample complexities of $\mathcal{O}(\epsilon^{-11})$ (Theorem 3) and $\mathcal{O}(\epsilon^{-17})$ (Theorem 6) respectively.  Further, the authors propose a zeroth-order method modificiation to warm-start deterministic MGDA in Algorithm 4, which uses only $O(1)$ space and time and achieves per-iteration $\epsilon$-level CA distance with same sample complexity $\mathcal{O}(\epsilon^{-11})$.

A key advantage of their analysis is removing the bounded gradient assumption required by all existing works and they require sufficient effort to show such a bound exists implicitly for their algorithms.


Finally, on two multi-task learning datasets (Cityscapes and NYU-v2), their warm start method obtains best average performance across all tasks.


**References** --
- (Zhang et al 2019) Why gradient clipping accelerates training: A theoretical justification for adaptivity. Arxiv.
- (Desideri 2012) Multiple-gradient descent algorithm (mgda) for multiobjective optimization. CRM.

**Strengths:**

- **Presentation**: The paper is easy to follow, even for readers without expertise in MOO. The MOO problem under generalized smoothness, the MGDA algorithm, and the warm-start procedure are well-motivated. Further, the proof sketch also provides a good overview of key ideas, atleast for average CA distance.


- **Novel algorithms**: A warm-start procedure appears in (Xiao et al 2023), however, it requires running two-loops, one of warm start of $w$ and other for $x$. In contrast, the proposed warm-start algorithm is run only once at initialization for $w$, and then the simple MGDA algorithm takes over. Even with this "weaker" warm-start, the authors can achieve stronger per-iteration $\epsilon$-CA distance. Algorithm 4, MGDA-FA, seems novel, and utilizes zeroth-order optimization intelligently, without worsening theoretical guarantees in the deterministic case.


- **Weak assumptions**: All existing works require $L$-smoothness and bounded gradients, while this paper requires only generalized smoothness. For their proofs, they show that gradients are indeed bounded by initial suboptimality, $\max_{i\in [K]} (f_i(x_0) - f_i^\star)$ for MGDA updates in the deterministic case. In the stochastic case, their proof is more complicated, wherein they take  a union bound over all iterations until a stopping time to show that gradient noise is also bounded.


- **Best sample complexity**: Even under weaker conditions, the obtained sample complexities for average CA distance, $\mathcal{O}(\epsilon^{-2})$ and $\mathcal{O}(\epsilon^{-4})$ for deterministic and stochastic cases respectively, match the lower bounds for single-objective optimization under $L$-smoothness. For iteration level CA distance, it seems that the best sample complexity of existing methods with stronger assumptions is $\mathcal{O}(\epsilon^{-12})$ while their analysis obtains $\mathcal{O}(\epsilon^{-17})$.

**Weaknesses:**

- **Theory**:
    - **Definition of $\mathcal{W}$:** The authors have not defined the set $\mathcal{W}$, however across the proof(Line 809, Eq 11), they use $\max_{w\in \mathcal{W}}\|w\| \leq 1$. It seems to be the unit sphere in $K$ dimensions.
    - **Choice of step size does not work in Theorem 2:** The parameters $\alpha,\beta,\rho = \mathcal{O}(\epsilon^2)$ do not work out for Theorem 2 as each of $\alpha, \beta$ and $\rho$ depend on other two. Consider $\rho \leq \frac{1}{\sqrt{\alpha T}}$ in Theorem 2, which for $T = \epsilon^{-4}$ and $\alpha  = \epsilon^{2}$ implies, $\rho \leq \epsilon$. For $\epsilon << 1$, $\rho=\epsilon^2$ does not satisfy this condition. Ideally, the authors should provide the condition for parameters in a sequential manner, for instance, $\rho$ depends on $\beta$ and $\alpha$, $\beta$ depends on $\alpha$ not $\rho$ and $\alpha$ doesn't depend on either $\beta$ or $\rho$. This would ensure that such inconsistent parameter values do not arise. Also, I am no longer sure that $T = \mathcal{O}(\epsilon^{-4})$ can be obtained for choice of $\alpha,\beta,\rho$ satisfying all constraints in Theorem 2.
    - **Value of $M$:**  For the deterministic case (Theorems 1, 3, 4 and 5), the value of step sizes depend on $M$ defined in Line 357, which is a function of $\ell$ and initial function suboptimality. Therefore, while the bounds in these theorems scale well with  $\epsilon$, they might not do so with $M$. If the scaling with $M$ in these theorems matches that of single-objective generalized $\ell$-smooth optimization from (Li et al 2024), or if is small under certain conditions on $\ell$, then it should not be an issue.
    - **Dependence on high probability error $\delta$ in stochastic variants (Theorem 2 and 6):** The authors do not provide the dependence on the high probability error $\delta$ in Theorems 2 and 6. For high probability guarantees, we would want to know if we can make $\delta$ extremely small, for instance $\mathcal{O}(\epsilon)$, or if it works only for a constant $\delta$. From Theorem 8, it seems that a constant $\delta \in (0,\frac{1}{2})$ might work, but the authors should provide more details on this. The union bound  over all iterations should force $\delta$ to not be very small.



- **Poor empirical Performance of MGDA-FA**: While MGDA-FA achieves the same performance as warm-start MGDA theoretically(Theorems 3 and 4), it performs much worse in experiments (Table 4). Its performance is better than only $2$ of the $8$ baselines in Table 3, while warm-start MGDA outperforms all $8$ baselines. The authors do not discuss possible causes for this. Even though MGDA-FA is fast in practice, as its error is very large, its usefulness is limited.




- **Typos**: $f_i^\star$ in the definition of $\Delta$ in Line 357 and "optional stopping theorem" in Line 407-408.

**Questions:**

- **References for Optimal rates for deterministic variants of per-iteration CA distance** : Can the authors provide any references for deterministic iteration-level CA distance to compare Theorems 3 and 4. Further, can the authors compare the performance of their warm-start procedure to existing baselines? Do warm-start procedures achieve optimal sample complexity for these cases? It might be helpful to include the sample complexity results in Table 1 for a quick comparison to baselines.

- **Independent samples for each $f_i$ for stochastic variants**: For
stochastic variants (Line 387) at each iteration $t$, $F(x_t;s_{t,i})$ requires a different sample $s_{t,i,j}$ to evaluate $f_j$ for $j\in [K]$. Is this necessary for the analysis? Does using the same sample for all $f_j$ change the analysis, as in practice, one might take a single minibatch to evaluate all $f_j$ at each iteration. Further, in Eq 16 and 17, using the same sample can remove the terms of $K$ from the union bound while still obtaining similar results.

- **Stochastic MGDA-FA** : If there is a bound the variance of $F(x;s)$ for instance $\mathbb{E}[\|F(x;s) - F(x)\|^2]\leq \sigma_F^2, \forall x$ or for each function $f_j(x;s)$, then would it be possible to analyze a stochastic variant of MGDA-FA by combining the analysis of the deterministic MGDA-FA and stochastic MGDA? Are there any reasons why the authors believe this shouldn't be possible or straightforward?

- **Warm-start procedure for general bi-level optimization**: MOO is bi-level optimization problem, where the inner optimization problem solves for best $w$ and the outer optimization solves for $x$. The warm-start procedure here implies that for single loop solutions to bilevel optimization, the inner problem is already very accurate, $\mathcal{O}(\epsilon)$ to be precise. This makes solving the bi-level optimization easier, as the inner problem is already almost solved.
Can this technique be applied to ohter bi-level optimization problems, where a single warm-start conveniently solves the inner problem accurately? Or is this an artifact of the MOO problem's simple structure?

---

> ### Author Response · Authors · 2024-11-21
>
> We thank the reviewer Wanu for the time and valuable feedback! Here are our responses.
>
> **Weakness 1:** Thanks for pointing it out. $\mathcal{W}$ is the probability simplex over $[K]$. We have moved the definition from section 2.3 to Definition 3 in the revision.
>
> **Weakness 2:** Thanks for the question. We believe that the selection in Theorem 2 is reasonable and here is the reason. Let us say we choose $\alpha, \beta, \rho=\epsilon^2$ and $T=\epsilon^{-4}$. Then the selection for $\rho$ becomes $\rho=\min\\{\mathcal{O}(\epsilon^2), \epsilon, \mathcal{O}(1), \mathcal{O}(\epsilon^2)\\}$. Then there is no contradiction since $\rho=\epsilon^2\ll\epsilon\ll 1$. We have noticed that the notations are not clear and we revise them as follows. Following your suggestion, we rewrite the selections for the parameters in Theorem 2: we choose $\rho\le \mathcal O(\epsilon^2), \beta\le \mathcal O(\min\\{\epsilon^2,\rho\\}), \alpha\le \mathcal O(\min\\{\beta, \rho, \frac{1}{\beta T}, \frac{1}{\rho T}\\})$ and $T\ge \max\\{\frac{1}{\alpha \epsilon^2},\frac{1}{\epsilon^4}\\}$ we can get the convergence. Thus we need to set $\alpha,\beta, \rho \le \mathcal O(\epsilon^2)$ and $T\ge \Theta(\epsilon^{-4})$. More details about the dependences on $M,\delta$ will be shown in the answer for **weakness 4**.
>
> **Weakness 3:** For the deterministic setting,  $M$ depends on the $\Delta$, which is the gap between the initial function value and optimal value. Since the loss function is typically non-negative, $\Delta$ is bounded by the initial function value and can be assumed as a constant. The scale of $M$ also depends on the $\ell$ function. We then show a fair comparison between our average CA distance methods (algorithms 1 and 4) with [a] when $\ell$ is a linear function. Then we have $M=\mathcal O(\Delta)$ and [a] shows a total sample complexity of $\mathcal{O}(M^2\epsilon^{-2})$. Our algorithm 1 requires $\mathcal{O}(M^3\epsilon^{-2})$. The additional complexity in our methods is introduced because we required smaller stepsize $\beta$ to update the CA direction while the CA direction is not needed in single-task setting. Our algorithm 4 requires $\mathcal{O}(M^5\epsilon^{-2})$ due to the employment of fast approximation.
> However, in the deterministic setting, $M$ is a constant. Thus it should not be an issue.
>
> [a] Li, H., Qian, J., Tian, Y., Rakhlin, A., \& Jadbabaie, A. (2024). Convex and non-convex optimization under generalized smoothness. Advances in Neural Information Processing Systems, 36.
>
> **Weakness 4:** Thanks for pointing this out.  We first rewrite the selections of parameters: $\rho\le \mathcal O(\delta^2 \epsilon^2), \beta\le \mathcal O(\min\\{\frac{\delta\epsilon^2}{M^4},\frac{\rho}{M^2}\\}), \alpha\le \mathcal O(\min\\{\beta, \frac{\rho}{\ell(M+1)^2},  \frac{\rho}{M^2}, \frac{\delta }{\rho T},\frac{1}{\beta TM^2}\\})$ and $T\ge\Theta (\max\\{\frac{1}{\delta \alpha \epsilon^2},\frac{M^2}{\delta^2 \epsilon^4}\\})$. The scale of $M,\ell(M+1)$ also depends on the $\ell$ function. We then show a fair comparison between our average CA distance method with [a] when $\ell$ is a linear function. For stochstic settting, we then have that $M,\ell(M+1)=\mathcal O(\frac{\Delta}{\delta})$. Thus the total sample complexity in [a] is $\mathcal O(\delta^{-4}\epsilon^{-4})$. By choosing $\rho\le \mathcal O(\delta^2\epsilon^2),\beta\le \mathcal O(\delta^5\epsilon^2),\alpha\le \mathcal O(\delta^5\epsilon^2) $ and $T\ge \Theta(\delta^{-6}\epsilon^{-4})$, our  sample complexity is $\mathcal O(\delta^{-6}\epsilon^{-4})$. The additional complexity in our methods is introduced because we required smaller stepsize $\beta$ to update the CA direction while the CA direction is not needed in single-task setting.
>
> **Weakness 5:** Good observation. We speculate that the fast approximation may introduce substantial errors in practice, potentially leading to inaccuracies in the weight update process. Meanwhile, the experiment uses the stochastic variant of MGDA-FA. According to our response to your **Question 3**, the convergence analysis is not straightforward under stochastic settings. However, it is premature to conclude that this approach has limited utility. This fast approximation comes from a similar idea in FAMO [b] and their performance is satisfying because they apply a logarithmic loss technique. We can also use this logarithmic technique to enhance our performance. We have added this discussion in Section B.2.
>
> [b] Liu, B., Feng, Y., Stone, P., \& Liu, Q. (2024). Famo: Fast adaptive multitask optimization. Advances in Neural Information Processing Systems, 36.
>
> **Weakness 6:** Thanks for pointing them out. We have revised all related typos.

---

> ### Author Response · Authors · 2024-11-21
>
> **Question 1:** Good question. To the best of our knowledge, existing studies on iteration-wise CA distance are conducted under stochastic settings, with SDMGrad being a representative work. We are not aware of any deterministic variants. Additionally, we have evaluated the performance of MoCo-warm start, and the results on both Cityscapes and NYU-v2 datasets are provided below. As shown in the tables, the warm start strategy consistently enhances performance. The warm start procedure requires very low sample complexity, as it computes the gradients only once and reuses them throughout the gradient descent. Thus, it does not affect the total sample complexity as the main loop dominates. The warm start process is introduced to achieve iteration-wise CA distance.
> | Method| Segmentation || Depth|| $\Delta m\% \downarrow$ |
> |-|-|-|-|-|-|
> || mIoU $\uparrow$ | Pix Acc $\uparrow$ | Abs Err $\downarrow$ | Rel Err $\downarrow$||
> | MoCo| 75.42| 93.55| 0.0149| 34.19 | 9.90 |
> | MoCo-warm start   | 75.48| 93.54 | 0.0148 | 31.43| 7.32|
>
> *Table: Multi-task supervised learning on Cityscapes dataset.*
>
> | Method| Segmentation||Depth|||Surface Normal|||| $\Delta m\% \downarrow$|
> |-|-|-|-|-|-|-|-|-|-|-|
> || mIoU $\uparrow$| Pix Acc $\uparrow$ | Abs Err $\downarrow$ | Rel Err $\downarrow$ | Angle Distance $\downarrow$ | Mean| Median     | Within $11.25^\circ$ $\uparrow$ | Within $22.5^\circ$ $\uparrow$ | Within $30^\circ$ $\uparrow$ |
> | MoCo| 40.30| 66.07| 0.5575| 0.2135| 26.67| 21.83| 25.61| 51.78| 64.85| 0.16|
> | MoCo-warm start| 38.40| 64.40| 0.5377| 0.2315| 26.04| 20.57| 27.11| 54.02| 66.63| -0.88|
>
> *Table: Multi-task supervised learning on NYU-v2 dataset.*
>
> Besides, thank you for your suggestion. We have incorporated the sample complexity in Table 1 with the most relevant papers, including MoCo, SDMGrad, and MoDo, as these are the only works that examine the CA distance. For other papers that do not explore the CA distance or rely on dissimilar assumptions, we have marked their sample complexity as N/A.
>
> **Question 2:** Thanks for pointing this out. It is OK to use the same sample but the union bound are similar and the term $K$ can not be moved. This is because in Eq 16 and 17 the term $K$ is introduced by the estimate error for each task. Even the tasks share the same sample, the estimate errors for each task are different thus the $K$ in the union bound is requried. The only difference is in Section C.4 that for any $t\le T$ and $i\in [3]$, the elements of $\varepsilon_{t,i}$ are dependent and it requires more efforts to bound this norm square. By using multiple samples, the elements of $\varepsilon_{t,i}$ are independent, which makes its norm square easy to bound.
>
> **Question 3:** Good question! We have checked the stochastic variant of MGDA-FA with bounded variance assumption. It turns out that there will be additional error terms that are not easy to control. We provide a brief analysis here following the same steps in Theorem 10 (Appendix). Since we have the bounded variance assumption now, we can derive
> \begin{align*}
> \mathbb{E}[\langle w_t-w, \nabla F(x_t)^\top\nabla F(x_t)w_t\rangle]\leq\frac{1}{2\beta}\mathbb{E}[\\|w_t-w\\|^2-\\|w_{t+1}-w\\|^2]+\rho+\frac{\alpha K\ell(M+1)M^2}{2}+\beta(C_1^\prime)^2+\frac{2\beta\sigma_F^2}{\alpha^2}+\frac{4\sigma_F}{\alpha}.
> \end{align*}
> Then the equation (34) will become
> \begin{align*}
> \mathbb{E}[F(x_{k+1})w-F(x_0)w]\leq\frac{\alpha}{\beta}+\alpha\rho T+\frac{\alpha^2K\ell(M+1)M^2T}{2}+\alpha\beta(C_1^\prime)^2T+\frac{2\beta\sigma_F^2 T}{\alpha}+4\sigma_F T.
> \end{align*}
> As a result, we cannot get the result that $f_i(x_{k+1})-f_i^*\leq F$ since we cannot control the additional terms. Meanwhile, if we use a large batch $n_s$, the above inequality would become
> \begin{align*}
> \mathbb{E}[F(x_{k+1})w-F(x_0)w]\leq\frac{\alpha}{\beta}+\alpha\rho T+\frac{\alpha^2K\ell(M+1)M^2T}{2}+\alpha\beta(C_1^\prime)^2T+\frac{2\beta\sigma_F^2 T}{\alpha n_s}+4\frac{\sigma_F T}{\sqrt{n_s}}.
> \end{align*}
> Then with an extremely large batch size $n_s=\mathcal{O}(T^2)$, it could converge but this selection is impractical and computationally consuming. Therefore, the analysis for the stochastic variant of MGDA-FA is not straightforward under generalized smoothness.
>
> **Question 4:** Excellent insight! A warm start can indeed facilitate bilevel optimization. In this context, the inner problem typically produces an iterative decay of $\\|y_t-y(x_t)^*\\|$, where $x_t, y_t$ are variables of the outer and inner problem, respectively. $y(x_t)^*$ represents the optimal point of the inner problem at the $t$-th epoch. If a warm start is added before the bi-level main loop, $\\|y_0-y(x_0)^*\\|$ is already in a small magnitude, which makes bi-level optimization easier. Furthermore, this warm start process is not a MOO problem's simple structure. It can be applied to other problems such as bilevel optimization.

---

> > ### Comment · Reviewer_Wanu · 2024-11-22
> > **Response to Author's Comments**
> >
> > I thank the authors for their detailed review and revision. All my questions have been satisfactorily answered, and I am satisfied with their answers. I made a mistake in my Weakness 2, which the authors have correctly pointed out in their response.

---

> > > ### Author Response · Authors · 2024-11-22
> > >
> > > Dear Reviewer Wanu,
> > >
> > > Thanks so much for your reply! We are happy that our responses address your concerns.
> > >
> > > Best Regards, Authors

---

### Author Response · Authors · 2024-11-21

Dear reviewers,

We sincerely thank all reviewers for their valuable comments and constructive feedback on our work. To address the concerns, we have included our responses and conducted additional experiments, which are now added to the revised manuscript. Furthermore, we have carefully revised the paper to incorporate the suggested improvements, with all revisions clearly marked in blue. We believe these updates have significantly strengthened the quality of our work. Thank you once again for your insightful suggestions and thoughtful review.

Best,
Authors

---

### Meta-Review · Area_Chair_wkXR · 2024-12-19

**Metareview:**

This paper analyzes the multi-objective optimization problem where the goal is to minimize the minimum of K different objectives. The standard measure for optimality in this case is called \epsilon-conflict-avoidant (CA). The main contribution of the paper is to extend existing analysis to the setting where the objectives are "general l-smooth", where the Hessian is bounded by a function of norm of gradient. For standard CA definition the result matches the earlier results that assume smoothness. The paper also considers a stronger version of CA and gets results with worse \epsilon dependency. The reviewers agree that the results are novel and the generalization to general smoothness is interesting. Most reviewers also find the empirical results convincing, although there are some concerns on the performance of MGDA-FA. Overall all reviewers are positive.

**Additional Comments On Reviewer Discussion:**

Reviewers are in full agreement.

---

### Decision · Program_Chairs · 2025-01-22

Accept (Poster)